# Type XVII collagen coordinates proliferation in the interfollicular epidermis

Mika Watanabe[1], Ken Natsuga[1]*, Wataru Nishie[1], Yasuaki Kobayashi[2], Giacomo Donati[3,4], Shotaro Suzuki[1], Yu Fujimura[1], Tadasuke Tsukiyama[5], Hideyuki Ujiie[1], Satoru Shinkuma[1,6], Hideki Nakamura[1], Masamoto Murakami[7], Michitaka Ozaki[8], Masaharu Nagayama[9], Fiona M Watt[3], Hiroshi Shimizu[1]*

[1]Department of Dermatology, Hokkaido University Graduate School of Medicine, Sapporo, Japan; [2]Center for Simulation Sciences, Ochanomizu University, Tokyo, Japan; [3]Centre for Stem Cells and Regenerative Medicine, King's College London, London, United Kingdom; [4]Department of Life Sciences and Systems Biology, University of Turin, Turin, Italy; [5]Department of Biochemistry, Hokkaido University Graduate School of Medicine, Sapporo, Japan; [6]Division of Dermatology, Niigata University Graduate School of Medical and Dental Sciences, Niigata, Japan; [7]Department of Dermatology, Ehime University Graduate School of Medicine, Toon, Japan; [8]Department of Biological Response and Regulation, Faculty of Health Sciences, Hokkaido University, Sapporo, Japan; [9]Research Institute for Electronic Science, Hokkaido University, Sapporo, Japan

*For correspondence: natsuga@ med.hokudai.ac.jp (KN); shimizu@ med.hokudai.ac.jp (HS)

**Abstract** Type XVII collagen (COL17) is a transmembrane protein located at the epidermal basement membrane zone. COL17 deficiency results in premature hair aging phenotypes and in junctional epidermolysis bullosa. Here, we show that COL17 plays a central role in regulating interfollicular epidermis (IFE) proliferation. Loss of COL17 leads to transient IFE hypertrophy in neonatal mice owing to aberrant Wnt signaling. The replenishment of COL17 in the neonatal epidermis of COL17-null mice reverses the proliferative IFE phenotype and the altered Wnt signaling. Physical aging abolishes membranous COL17 in IFE basal cells because of inactive atypical protein kinase C signaling and also induces epidermal hyperproliferation. The overexpression of human COL17 in aged mouse epidermis suppresses IFE hypertrophy. These findings demonstrate that COL17 governs IFE proliferation of neonatal and aged skin in distinct ways. Our study indicates that COL17 could be an important target of anti-aging strategies in the skin.

## Introduction

Skin is a highly structured organ in which stem cell self-renewal, cell proliferation and differentiation are coordinated to maintain homeostasis. In addition to hair follicles and other skin appendages, the interfollicular epidermis (IFE; non-haired skin) comprises distinct cellular populations. IFE stem cells reside in the basal cell layer; these cells both self-renew and generate the terminally differentiated outer cell layers which function as barriers to the external environment and prevent loss of body fluids (*Giangreco et al., 2008*; *Hsu et al., 2014*; *Jones et al., 2007*; *Natsuga, 2014*).

Organismal aging, or physical aging, is defined as tissue impairment arising from the accumulation of numerous intrinsic and extrinsic factors that induce cell damage chronologically. The

**eLife digest** The skin is one of the largest organs of the body and is constantly confronted with a range of external stresses including germs, heat and scratches. The outermost part of the skin is called the epidermis and it acts as a barrier to the external environment and works to stop the body from losing water. An abnormally thin or thick epidermis can impair the skin's ability to perform these roles. As such, the ability of epidermal cells to proliferate (i.e. divide to make new cells) is tightly regulated, both when the animal first develops and when it ages. However, most of the underlying mechanisms that regulate these processes are unknown.

Watanabe et al. have now identified type XVII collagen (called COL17 for short) as a key molecule that controls how often epidermal cells in skin from mice and humans divide. COL17 is a protein that is made in the deepest layer of the epidermis, and it prevents the epidermis from thickening in newborn mice by coordinating with the Wnt signaling pathway. This signaling pathway, amongst other things, controls how often some cells divide.

Older mice have a thicker epidermis than their younger counterparts. Watanabe et al. revealed that the distribution of COL17 in the epidermis also changes dramatically with age in mice and humans. Further experiments with mice showed that introducing COL17 back into the epidermis helped the tissue retain a more youthful state even in animals that had reached an old age.

Together these findings give scientists a better understanding of how the ability of epidermal cells to divide is regulated at various stages in a mammal's life. The new findings also point to COL17 as a promising component in future anti-aging strategies targeted at the skin. Yet first, further work will be needed to uncover how the production of COL17 is controlled in the epidermis.

relationship between organismal aging and stem cells is an inescapable bond, and the heterogeneity of stem cells in organs may be reduced with aging (*Goodell and Rando, 2015*). Human skin aging is exemplified by alterations in the dermis and skin appendages, such as the thinning of dermis, dryness, wrinkles, gray hair and hair loss (*Rittié and Fisher, 2015*). However, the influence of aging on IFE has been controversial. Although decreased epidermal proliferation has been reported in in vitro and in vivo studies using aged individuals and mice (*Giangreco et al., 2008*; *Gilchrest, 1983*; *Grove and Kligman, 1983*), several recent studies have reported contradictory results, showing sustained and increased proliferation in the aged epidermis (*Charruyer et al., 2009*; *Stern and Bickenbach, 2007*). Thus, how organismal aging affects the IFE and its stem cells has not been clarified (*Keyes et al., 2013*).

The extracellular matrix proteins of the basement membrane zone (BMZ) are important components of the IFE stem cell niche and connect the dermis and epidermis functionally. Type XVII collagen (COL17) is a type II transmembrane protein that is located along the hemidesmosomes in the BMZ. The N-terminus of COL17 is localized in hemidesmosomes, and its extracellular domain reaches the lamina densa (*McMillan et al., 2003*). Non-hemidesmosomal COL17 in keratinocytes and human skin has also been reported (*Hirako et al., 1998*). COL17 has been characterized as a target protein in the autoimmune blistering disease bullous pemphigoid (*Nishie, 2014*) and also as being the defective protein in junctional epidermolysis bullosa (JEB), a congenital blistering disease (*Fine et al., 2014*). Recently, COL17 has been shown to form a niche for hair follicle stem cells (HFSCs), as mice lacking the protein and human JEB patients with mutations in *COL17A1*, the open-reading frame encoding COL17, exhibit a premature aged skin phenotype, including gray hair and hair loss (*Matsumura et al., 2016*; *Nishie et al., 2007*; *Tanimura et al., 2011*). Additionally, reduced labeling of epidermal basement membrane proteins, including COL17, in human skin has been associated with aging (*Langton et al., 2016*). However, the role of COL17 in maintaining IFE and its stem cells is still unclear.

In the present study, we explore the comprehensive role of COL17 in regulating IFE homeostasis and also characterize age-related IFE alterations associated with a modified BMZ, including COL17. We show that COL17 is indispensable for regulating IFE proliferation in neonatal mice through activating the Wnt pathway. The IFE hyperproliferation is induced by organismal aging and can be reversed by COL17 replenishment.

## Results

### COL17 deficiency leads to epidermal hyperproliferation in neonatal IFE

We investigated the phenotype of paw skin to study IFE in *Col1a1−/−* mice (*Nishie et al., 2007*) in the absence of hair follicles. Neonatal IFE skin (P1, postnatal day 1) of *Col1a1−/−* mice showed transient epidermal hyperproliferation, as demonstrated by counting the epidermal layers and the

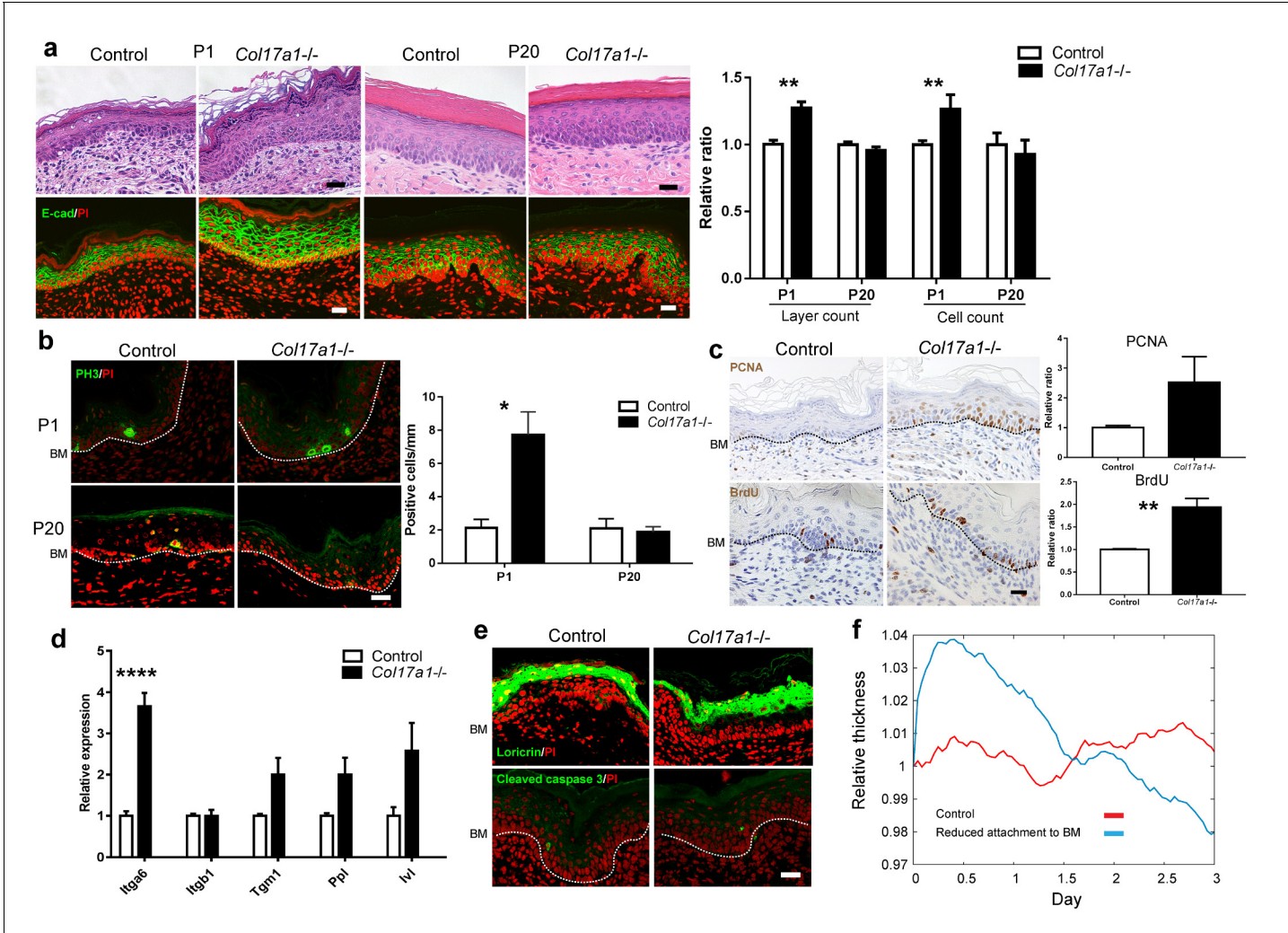

**Figure 1.** COL17 deletion induces transient IFE hyperproliferation in neonates. (**a**) Hematoxylin and eosin (H&E) staining and E-cadherin (E-cad) labeling (with PI nuclear counterstain) of *Col1a1−/−* and control IFE skin samples from *Col1a1+/-* or *Col1a1+/+* littermates (Control) at P1 (n = 5) and P20 (n = 4). Scale bar: 20 μm. Quantitation of the number of epidermal layers and epidermal cell counts. The values are shown as relative ratios to the controls. (**b**) PH3 staining at P1 and P20. Scale bar: 20 μm. The number of epidermal basal cells positively labeled for PH3 per mm epidermis (n = 4). BM, basement membrane. (**c**) PCNA and BrdU labeling at P1. Scale bar: 20 μm. Quantitation of PCNA- (n = 5) and BrdU-positive basal cells (n = 4). The values are shown as relative ratios to the controls. (**d**) Quantitative RT-PCR (qRT-PCR) of *Itga6*, *Itgb1*, *Tgm1*, *Ppl* and *Ivl* mRNAs (n = 5). (**e**) Loricrin and cleaved caspase-3 staining (representative images from 3 mice). Scale bar: 20 μm. BM, basement membrane. (**f**) An in silico model of the epidermal cell proliferation upon the reduced adhesion of committed progenitor cells to the BMZ. The details are described in the Material and Methods. The data in all of the histograms are the means ± SE. *0.01<p<0.05, **0.001<p<0.01, ****p<0.0001. Student's t-tests.

The following figure supplements are available for figure 1:

**Figure supplement 1.** Barrier function assay of *Col1a1−/−* and littermate controls (E18.5).

**Figure supplement 2.** Proliferative ability of the back skin IFE from *Col1a1−/−* mice and NHEKs treated with *COL17A1* siRNAs.

numbers of epidermal cells and phospho-Histone H3 (PH3)-positive cells (*Figure 1a–b*). The numbers of proliferating cell nuclear antigen (PCNA)- and Bromodeoxyuridine (BrdU)-positive cells in the *Col17a1−/−* IFE basal cells at P1 were also increased compared with the controls (*Figure 1c*), indicating that COL17 deletion affects both the S and M phases in the cell cycle of IFE neonatal keratinocytes. The proliferative IFE phenotype of *Col17a1−/−* mice gradually waned postnatally, and the epidermal thickness and the number of PH3-positive cells became comparable with those of the controls at P20 (*Figure 1a–b*).

We investigated whether the expression levels of markers of basal cells and differentiated cells were altered in the hyperproliferative IFE of *Col17a1−/−* mice at P1. The gene expression of *Itga6*, which might compensate for COL17 deficiency, was increased in *Col17a1−/−* IFE skin, whereas the expression of *Itgb1, itgb4, lamb3* and *lamc2* was not altered (*Figure 1d*, *Figure 1—figure supplement 1a*). The mRNA expression levels of *Tgm1, Ppl* and *Ivl* were somewhat higher in *Col17a1−/−* IFE skin (*Figure 1d*), while loricrin-labeled granular cell layers were not expanded in these mice (*Figure 1e*). Dye-permeability and transepidermal water loss at day 18.5 of embryogenesis (E18.5) were comparable between the *Col17a1−/−* mice and the controls (*Figure 1—figure supplement 1b–c*), indicating that IFE keratinocyte differentiation was not greatly altered in *Col17a1−/−* skin. There was no increase in the number of cells positive for cleaved caspase-3, a marker of apoptosis, in the *Col17a1−/−* epidermis compared with the controls (*Figure 1e*). The electron microscopy results indicated that the dermo-epidermal junction of the *Col17a1−/−* IFE presented hypoplastic hemidesmosomes in accordance with previous observations on the back skin of *Col17a1−/−* mice (*Nishie et al., 2007*) (*Figure 1—figure supplement 1d*). There were no significant differences in the number of inflammatory infiltrates, including CD3+, F4/80+ and Ly-6G+ cells, in the dermis of *Col17a1−/−* mice and control mice, which excludes inflammation as a contributor to immature hemidesmosome formation (*Figure 1—figure supplement 1e–g*).

To explain the transient epidermal hyperproliferation of the *Col17a1−/−* neonatal IFE, we exploited an in silico model (*Kobayashi et al., 2016*) to recapitulate the epidermal development. As COL17 serves as a linker between the epidermis and dermis and is expressed in basal cells, most of which are committed progenitor cells in the epidermis (*Doupé and Jones, 2012*; *Lim et al., 2013*), epidermal thickness was calculated upon the loosening of the attachment of committed progenitor cells to the BMZ. In the model with diminished adhesion, the epidermal thickness was transiently increased and gradually returned to the baseline (*Figure 1f*), which was compatible with the experimental observations of *Col17a1−/−* neonates and with the transient hypertrophy of *Itga6*- and *Itgb1*-null epidermis (*Brakebusch et al., 2000*; *Niculescu et al., 2011*). These data suggest that COL17 deletion induces epidermal thickening by the hyperproliferation of IFE keratinocytes, at least partially through the loosening of dermo-epidermal adhesion at the neonatal stage.

The site specificity of the hyperproliferative phenotype in the neonatal paw epidermis of *Col17a1/-/* mice was confirmed by the comparable expression of proliferation markers in the back skin IFE of *Col17a1−/−* mice and control mice (*Figure 1—figure supplement 2a*). The discordance between the paw epidermis and back skin IFE might be explained either by the influence of hair follicle development on the back skin IFE or by the distinct regulation of the IFE at each body site (*Rompolas et al., 2016*; *Roy et al., 2016*; *Sada et al., 2016*). We also investigated cell-intrinsic properties due to COL17 defects using cultured normal human epidermal keratinocytes (NHEKs). The cell proliferation rates of NHEKs treated with *COL17A1* siRNAs were slightly decreased (*Figure 1—figure supplement 2b–d*), which is compatible with reduced proliferation of cultured keratinocytes derived from *Col17a1−/−* mice (*Tanimura et al., 2011*), and the colony-forming abilities of these cells were similar to those of control cells (*Figure 1—figure supplement 2e–f*). These data indicate that the proliferation potential of *Col17a1−/−* IFE is dependent on in vivo conditions.

## COL17 regulates neonatal IFE proliferation through Wnt-β-catenin signaling

Various signaling molecules are involved in controlling hair follicle stem cells and epidermal homeostasis (*Kretzschmar and Watt, 2014*); the relationships between these signaling molecules and BMZ proteins are only partially understood (*Margadant et al., 2010*; *Rognoni et al., 2014*; *Tanimura et al., 2011*). To further explore the underlying mechanisms of the transient epidermal hyperproliferation phenotype of *Col17a1−/−* IFE, we first screened the gene expression profiles of the receptors, co-receptors, transcription factors and cofactors of the major signaling pathways,

including Wnt, TGF-$\beta$/BMP, Notch, Hedgehog and FGF in neonatal *Col17a1−/−* and control skin samples. The screening data showed that the expression levels of Wnt-related molecules (*Fzd4*, *Nfatc2*, *Nfatc4* and *Tcf7*) were significantly decreased in *Col17a1−/−* neonatal IFE skin compared with controls (*Figure 2—figure supplement 1a*). Although the expression of some TGF-$\beta$−related genes was altered in *Col17a1−/−* IFE skin (*Figure 2—figure supplement 1a*), the TGF-$\beta$ and p-Smad2 immunostaining in *Col17a1−/−* IFE keratinocytes was comparable with that in IFE cells in the controls, in contrast to the TGF-$\beta$ and p-Smad2 reduction in *Col17a1−/−* HFSCs (*Tanimura et al., 2011*) (*Figure 2—figure supplement 1b–c*). The gene expression profiles for Notch, Hedgehog and the FGF pathway were not significantly affected upon COL17 deletion (*Figure 2—figure supplement 1a*).

To validate these findings, we analyzed the gene expression profiles of specific Wnt-related molecules in the independent replication samples. In line with the screening qPCR data, the expression levels of the Wnt target genes (*Tcf7 l1*, *Tcf7 l2*, and *Axin2*), the receptor gene (*Fzd4*), and stimulatory Wnt genes (*Wnt2*, *Wnt2b*, *Wnt5a*) were significantly downregulated in *Col17a1−/−*mice, whereas expression of the inhibitory Wnt gene (*Wnt4*) was increased (*Figure 2a*) (*Bernard et al., 2008*; *Mikels and Nusse, 2006*). The changes in these genes were paralleled by a reduction of basal cells positive for LEF1, a nuclear Wnt mediator of Wnt signaling, in neonatal *Col17a1−/−* mice IFE (*Figure 2b*). This reduction was also transient, and the number of LEF1-positive basal cells was comparable between the *Col17a1−/−* mice and the control mice at P4, which might account for the transient epidermal hyperproliferation of the paw IFE in *Col17a1−/−* mice. As $\beta$-catenin binds to LEF1 in the nucleus in the Wnt-canonical pathway (*Lien and Fuchs, 2014*), we performed $\beta$-catenin immunostaining. The number of nuclear $\beta$-catenin-positive cells in *Col17a1−/−* mice IFE was diminished compared to the controls (*Figure 2c*).

To further evaluate the relationship between Wnt signaling and COL17, we utilized SuperTop-Flash 293 (STF293) reporter cells (*Tsukiyama et al., 2015*). Overexpression of human COL17 significantly upregulated Wnt activity in this cell line (*Figure 2d*). Additionally, to visualize Wnt signaling in vivo, we crossed ins-Topgal+ mice (*Moriyama et al., 2007*) with *Col17a1−/−* mice. The LacZ-positive area that was indicative of active Wnt signaling in the IFE was significantly diminished in the ins-Topgal+:*Col17a1−/−* mice (*Figure 2e*, *Figure 2—figure supplement 2*). These results suggest that COL17 expression stabilizes Wnt signaling.

To examine whether these findings correlate with the phenotype of JEB patients with COL17 deficiency, we also performed immunostainings for LEF1, $\beta$-catenin and PH3 in JEB skin. In the JEB epidermis, the numbers of LEF1-positive cells and cells with nuclear $\beta$-catenin were decreased, while the number of PH3-positive cells was elevated (*Figure 2—figure supplement 3*); these findings were compatible with the data from the *Col17a1−/−* mice, although this result requires further investigation due to the small sample size.

We next investigated whether a defect in Wnt signaling in the IFE could be responsible for the hyperproliferation phenotype and whether that phenotype could be reversed by the introduction of COL17. K14-deltaNLef1 mice express a *Lef1* transgene that lacks a $\beta$-catenin-binding site under the control of the keratin 14 (K14) promoter and serve as a model of inactive Wnt signaling in the epidermis (*Niemann et al., 2002*). Neonatal K14-deltaNLef1 mice IFE exhibited epidermal thickening and a larger number of PH3-positive basal cells than the controls (*Figure 2f*). The number of PCNA-positive basal cells was also increased, albeit not significantly. To confirm that the abated Wnt activities were related to neonatal IFE proliferation in vivo, we intraperitoneally administered Wnt inhibitors (IWP-2 or Wnt-C59) into wild-type mouse neonates (*Kuo et al., 2016*; *Carotenuto et al., 2017*). The number of BrdU- and PH3-positive epidermal cells was increased in mice treated with these inhibitors at P1 compared with untreated control mice (*Figure 2g*). Transgenic rescue by the expression of human COL17 (hCOL17) under the K14 promoter in *Col17a1−/−* mice (*Nishie et al., 2007*) abrogated IFE hyperproliferation and restored LEF1- and nuclear $\beta$-catenin-positive basal cells (*Figure 3a,c–d*). The expression levels of Wnt-related genes that were altered in *Col17a1−/−* mice at P1 were restored by transgenic rescue with hCOL17 (*Figure 3b*). These data demonstrate that COL17 unambiguously contributes to the maintenance of neonatal IFE proliferation via its effect on Wnt signaling.

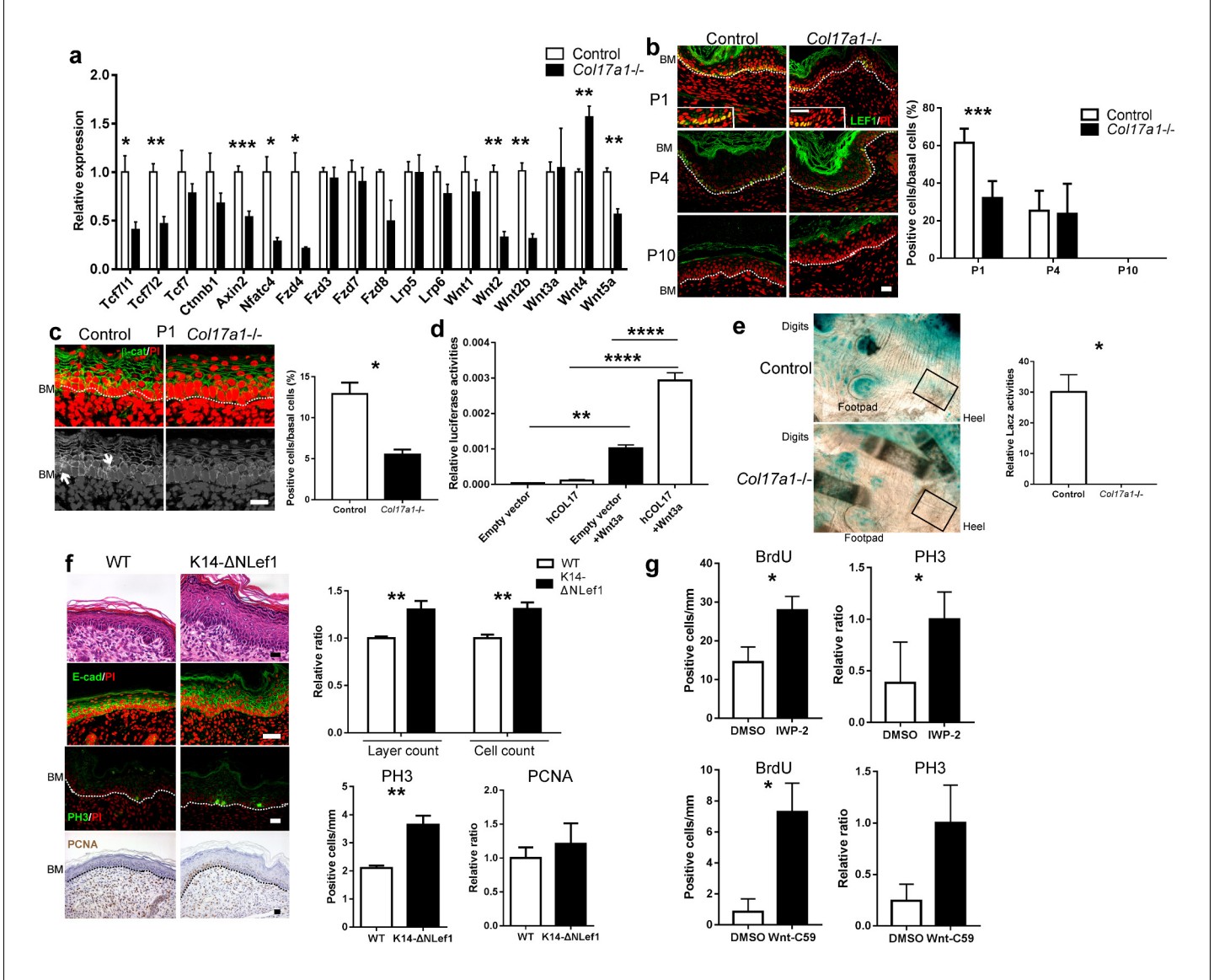

**Figure 2.** COL17 deficiency destabilizes Wnt-β-catenin signaling in neonates. (**a**) qRT-PCR of Wnt-related molecules in *Col17a1−/−* and control IFE skin samples at P1 (n = 5). Student's t-test. (**b**) LEF1 staining of *Col17a1−/−* and control IFE skin at P1 (n = 5), P4 (n = 4) and P10 (n = 4). Inlet: higher magnification of LEF1-positive basal cells at P1. Scale bar: 20 μm. Quantitation of LEF1-positive basal cells as a percentage of all basal cells (%). Student's t-test. (**c**) β-catenin staining of *Col17a1−/−* and control IFE skin at P1. Nuclear β-catenin accumulation is indicated with arrows. The quantification of nuclear β-catenin-positive cells (n = 3). Student's t-test. Scale bar: 20 μm. (**d**) Wnt activity in STF293 cells expressing hCOL17 treated with Wnt3a CM (n = 3). One-way ANOVA test, followed by Tukey's test. (**e**) Wnt activities in the hindpaw IFE from ins-Topgal+ (Control: left) and ins-Topgal+:*Col17a1−/−* (*Col17a1−/−*: right) mice. Calculated areas devoid of hair follicles or sweat glands are indicated with squares in the representative figures. The results are quantified as the Wnt-activated area per unit (n = 4). Scale bar: 100 μm. Mann-Whitney test. (**f**) H&E, E-cad, PH3 and PCNA staining of IFE skin samples from K14-ΔNLef and littermate controls at P1. Scale bar: 20 μm. The numbers of epidermal layers, epidermal cell counts and PCNA- and PH3-positive basal cells (n = 4). Student's t-test. (**g**) Quantification of BrdU- and PH3-positive cells in WT paw skin IFE treated with Wnt inhibitors (IWP-2 (n = 6) vs DMSO (n = 5) or Wnt-C59 (n = 6) vs DMSO (n = 4)). The data are presented as the means ± SE. Student's t-test. *0.01<p<0.05, **0.001<p<0.01, ***0.0001<p<0.001, ****p<0.0001.

The following figure supplements are available for figure 2:

**Figure supplement 1.** mRNA profiles of signaling molecules and TGF-β staining.

**Figure supplement 2.** The details of LacZ staining in ins-Topgal+ mice with COL17 deficiency.

*Figure 2 continued on next page*

*Figure 2 continued*

**Figure supplement 3.** Wnt signaling and proliferation profile in JEB patients epidermis with COL17 deficiency.

## Distribution of COL17 is altered with physical aging

Because *Col17a1−/−* mice (3-month-old) clearly exhibited the premature aging phenotypes of gray hair and hair loss (*Figure 4—figure supplement 1*) (*Nishie et al., 2007*; *Tanimura et al., 2011*), we examined the effects of physical aging on the IFE and the relationship between aging and COL17. We compared wild-type (WT) young mice (6 to 10 weeks old) with aged mice (19 to 27 months old). In the IFE of aged mice, the epidermis was thickened, and the numbers of PH3-, BrdU- and PCNA-positive cells were increased (*Figure 4a*). This phenomenon was specific to paw skin and was not observed in the back skin IFE (*Figure 4—figure supplement 2*), as was the case for the neonatal *Col17a1−/−* IFE. These data indicate that physical aging leads to paw IFE hyperproliferation.

The gene expression levels of *Itga6* and *Itgb1* were decreased in the aged mouse IFE, whereas that of *Col17a1* did not change (*Figure 4b*). Furthermore, the expression levels of differentiation markers (*Tgm1*, *Ppl*, and *Evpl*) were not greatly affected. Although *Col17a1* gene expression was unaffected, the distribution of COL17 was dramatically altered in the IFE of aged mice and humans. The apico-lateral portion of COL17 (non-hemidesmosomal COL17) was greatly reduced in the IFE of aged mice and humans, as detected with antibodies targeting the extracellular portion of COL17 (*Figure 4c*) and is in line with the recent publication on age-related alterations of COL17 in human

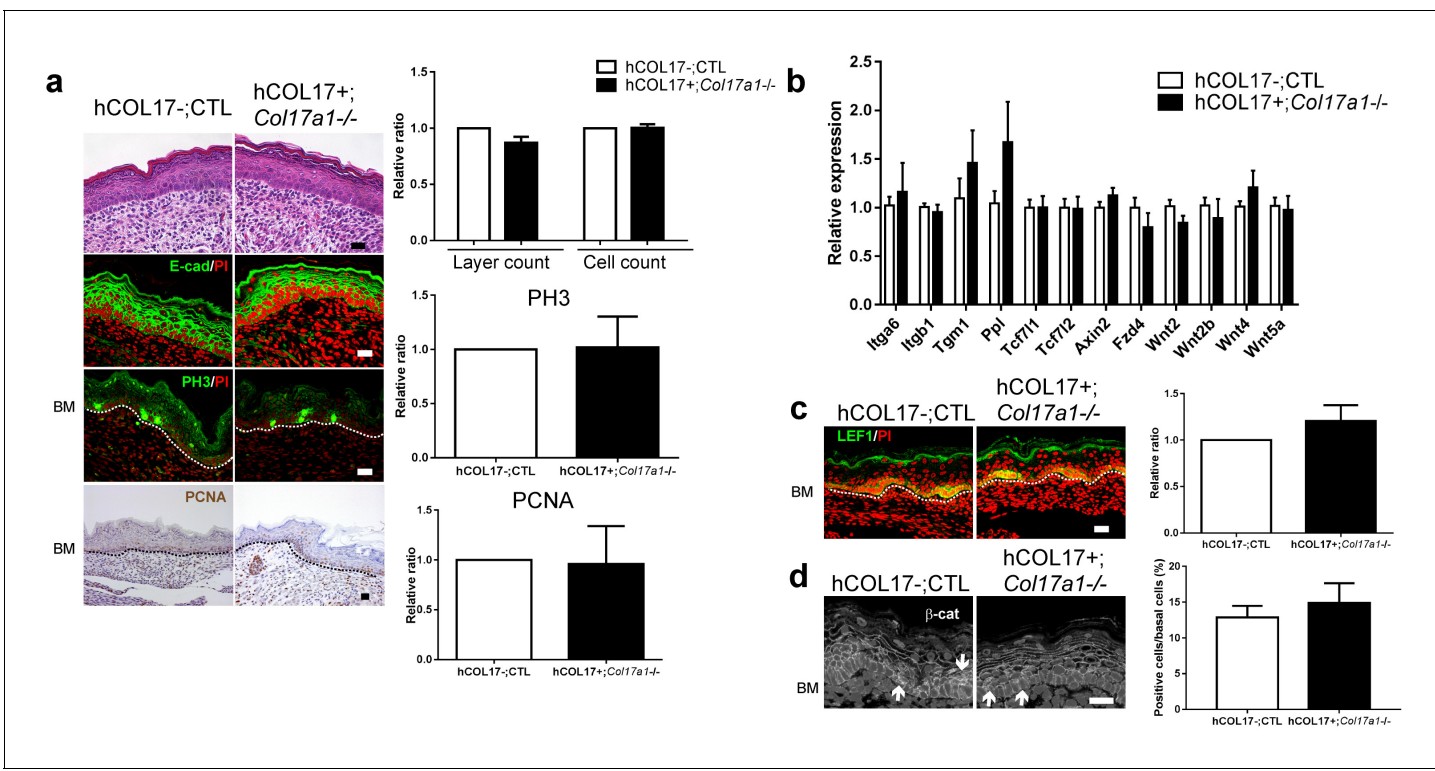

**Figure 3.** Induction of human COL17 abrogates epidermal hyperproliferation and the expression of Wnt-β-catenin signaling molecules in neonatal *Col17a1−/−* IFE. (a) H&E, E-cad, PH3 and PCNA staining of IFE skin specimens from *Col17a1+/+* or *Col17a1+/-* (as hCOL17-; CTL (control)) and hCOL17+; *Col17a1−/−* littermate mice at P1. Quantification of epidermal layers, epidermal cell counts, and PH3- and PCNA-positive cells (n = 4). Scale bar: 20 μm. (b) Gene expression of Wnt-related molecules in IFE skin samples from hCOL17-; CTL and hCOL17+; *Col17a1−/−* littermate mice at P1 (n = 4). (c) LEF1 staining of IFE skin samples from hCOL17-; CTL and hCOL17+; *Col17a1−/−* littermates at P1 (n = 4). Scale bar: 20 μm. (d) β-catenin staining of IFE skin samples from hCOL17-; CTL and hCOL17+; *Col17a1−/−* littermates at P1 (n = 4). Nuclear β-catenin is indicated with arrows. The number of nuclear β-catenin-positive cells. Scale bar: 20 μm. The data are the means ± SE. Student's t-tests.

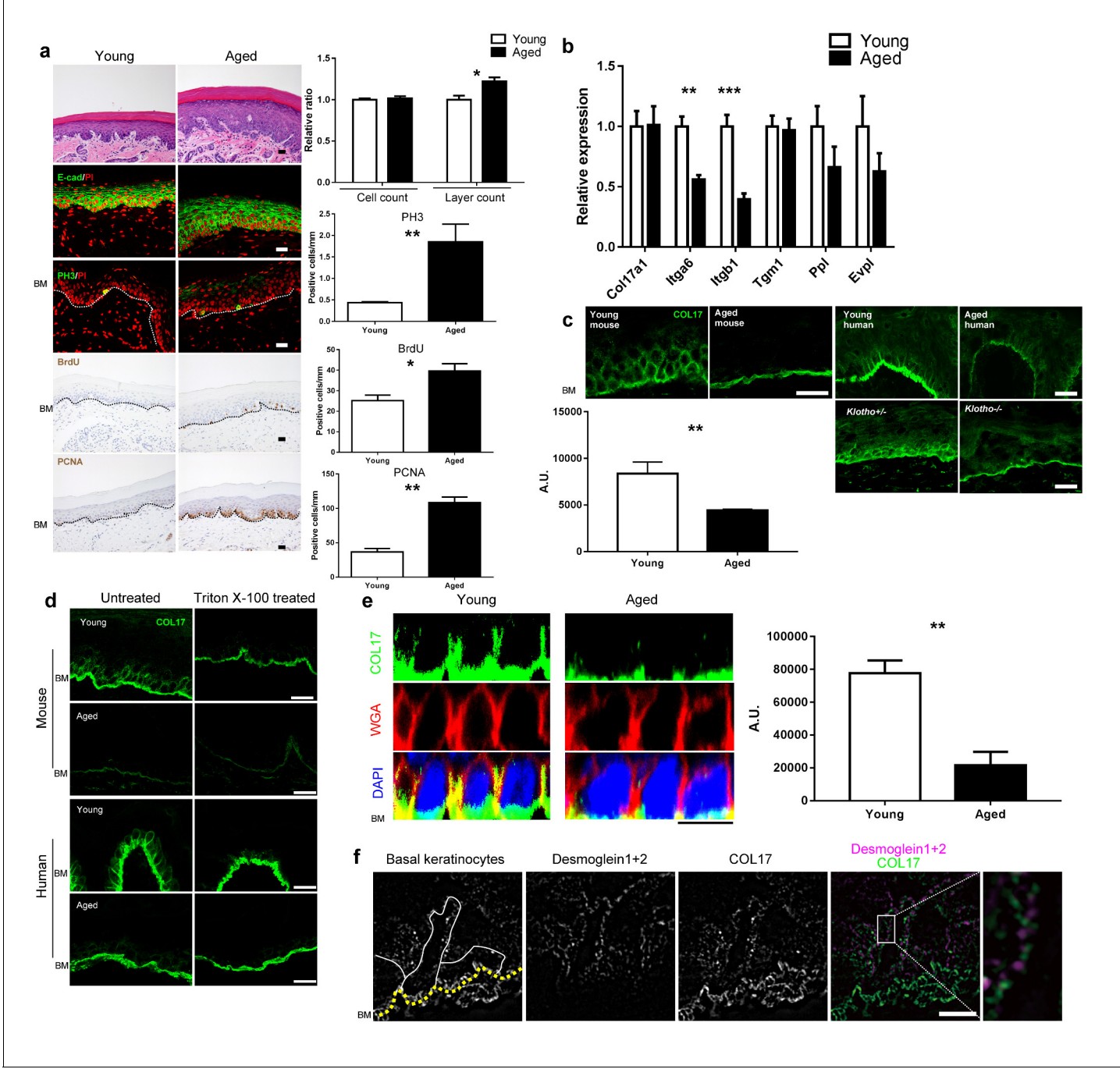

**Figure 4.** Physical aging affects epidermal proliferation and COL17 distribution. (a) H&E, E-cad, PH3, BrdU and PCNA staining of IFE skin from young (6–10 weeks old) and aged (19–27 months old) adult C57BL/6 wild-type (WT) mice. Scale bar: 20 µm. The numbers of epidermal layers, epidermal cell counts, and PH3-, BrdU- and PCNA-positive basal cells (n = 5). Student's t-test. (b) The gene expression levels of *Itga6*, *Itgb1*, *Tgm1*, *Ppl*, *Evpl*, and *Col17a1* in IFE skin samples from young and aged WT mice (n = 5 for *Itga6*, *Itgb1*, *Tgm1* and *Col17a1*; n = 3 for *Ppl* and *Evpl*). Student's t-test. (c) COL17 staining (antibodies to the juxtamembranous portion) in IFE skin samples from the following groups: young and aged WT mice (n = 5), young (<15 years old) and aged (>85 years old) normal human individuals (representative images from three human samples), and *Klotho+/-* and *Klotho−/−* littermates at 6 weeks (representative images from three mice). Scale bar: 20 µm. The quantitative fluorescent intensity of lateral membrane of IFE basal cells from young and aged WT mice (n = 5). Mann-Whitney test. (d) COL17 labeling following the Triton X-100 treatment of IFE skin from young/aged WT mice and human individuals (representative images from three samples). Scale bar: 20 µm. (e) The optical sectioning of 3D reconstructed whole mount COL17-stained skin from young and aged murine WT IFE. The IFE cell membrane was visualized with wheat germ agglutinin (WGA). DAPI (4′,6-diamidino-2-phenylindole) was used for nuclear staining. Scale bar: 10 µm. The quantitative fluorescent intensity of COL17 in lateral membrane of IFE basal cells from young and aged WT mice (n = 6). Mann-Whitney test. (f) The distributions of COL17 and desmogleins 1 and 2 in young murine WT IFE

*Figure 4 continued on next page*

*Figure 4 continued*

using N-SIM (structured illumination microscopy) image reconstruction (representative images from two mice). Basal keratinocytes were depicted by white lines. Scale bar: 5 μm. BM, basement membrane. The data are the means ± SE. *0.01<p<0.05, **0.001<p<0.01, ***0.0001<p<0.001.

The following figure supplements are available for figure 4:

**Figure supplement 1.** Gross appearance of *Col17a1−/−* and littermate controls (3 months old).

**Figure supplement 2.** Proliferation markers in back skin from young and aged mice.

**Figure supplement 3.** Intracellular COL17 labeling of the IFE from young/aged WT mice and human individuals.

**Figure supplement 4.** Expression of BMZ and extracellular matrix proteins in the IFE skin of young and aged WT mice and healthy individuals.

**Figure supplement 5.** Analysis of epidermal proteases in IFE skin samples from young and aged WT mice.

skin (*Langton et al., 2016*). This phenomenon was also observed in the IFE of *Klotho−/−* mice, a premature aging model (*Kuro-o, 2008*). The involvement of COL17 proteolytic cleavage, so-called ectodomain shedding (*Nishie et al., 2012*, *2015*), in the altered COL17 distribution was excluded using several antibodies that recognize the intracellular portions of COL17 (*Figure 4—figure supplement 3*).

To confirm the depletion of non-hemidesmosomal COL17 in the aged IFE, the sections were treated with Triton X-100 prior to immunostaining (*Hirako et al., 1998*). In both mouse and human skin, Triton X-100-treated young IFE lost the apico-lateral portion of COL17 and resembled the aged epidermis (*Figure 4d*). To further examine the reduction in the apico-lateral portion of COL17 in aged mice, we performed whole-mount staining of the IFE (*Figure 4e*). In 3D-reconstructed images, COL17 was confined to the basement membrane in the aged IFE, while basal cells of young mice had COL17 that was associated with the apical, lateral and basal membranes. Using high-resolution structured illumination microscopy (SIM) imaging on young WT mouse IFE (*Figure 4f*), COL17 was detected at the apico-lateral cell periphery of basal keratinocytes but did not co-localize with desmosomal proteins (desmogleins 1 and 2), indicating that the apico-lateral COL17 in basal keratinocytes was not incorporated into desmosomes, which are highly insoluble. In contrast to COL17 dynamics with aging, the distributions of integrins α6 and β1 were not modified by aging (*Figure 4—figure supplement 4a*). Other hemidesmosomal proteins (BP230 and Plectin) were also unchanged (*Figure 4—figure supplement 4b*). Sustained *Col17a1* gene expression in IFE skin contrasted with the decreased expression of many collagens and laminins with aging (*Figure 4—figure supplement 4c–d*). These results suggest that physical aging leads to paw IFE hypertrophy and modifies COL17 distribution in the epidermis in a posttranslational manner.

## Atypical PKC controls the distribution of COL17 in IFE

To elucidate the mechanisms that underlie the altered COL17 distribution that occurs with physical aging, we reproduced the environment of aged IFE. Because calcium distribution is changed in the aged epidermis (*Denda et al., 2003*) and the calcium concentration in the epidermis is lower in aged individuals than in young individuals (*Rinnerthaler et al., 2015*), whole IFE skin samples collected from young mice were treated with EDTA to simulate changes in calcium concentration in aged IFE. Whole mount staining showed that EDTA treatment eliminated apico-lateral COL17 in basal cells, while hemidesmosomal COL17 was present (*Figure 5a*). This finding suggests that the calcium concentration (or the concentrations of other molecules chelated by EDTA) may affect the COL17 distribution in the IFE.

Among the numerous cellular events influenced by calcium dynamics, we focused on atypical protein kinase C (aPKC), a key regulator of epithelial polarity (*Niessen et al., 2013*). The expression levels of aPKCζ and aPKCλ/ι, aPKC isoforms expressed by basal epidermal cells are dependent on the calcium concentration in cultured keratinocytes (*Helfrich et al., 2007*), and the ablation of aPKCλ/ι in the epidermis leads to premature aging phenotypes, such as gray hair, hair loss and IFE thickening, closely resembling *Col17a1−/−* mice (*Niessen et al., 2013*; *Osada et al., 2015*). Phospho-

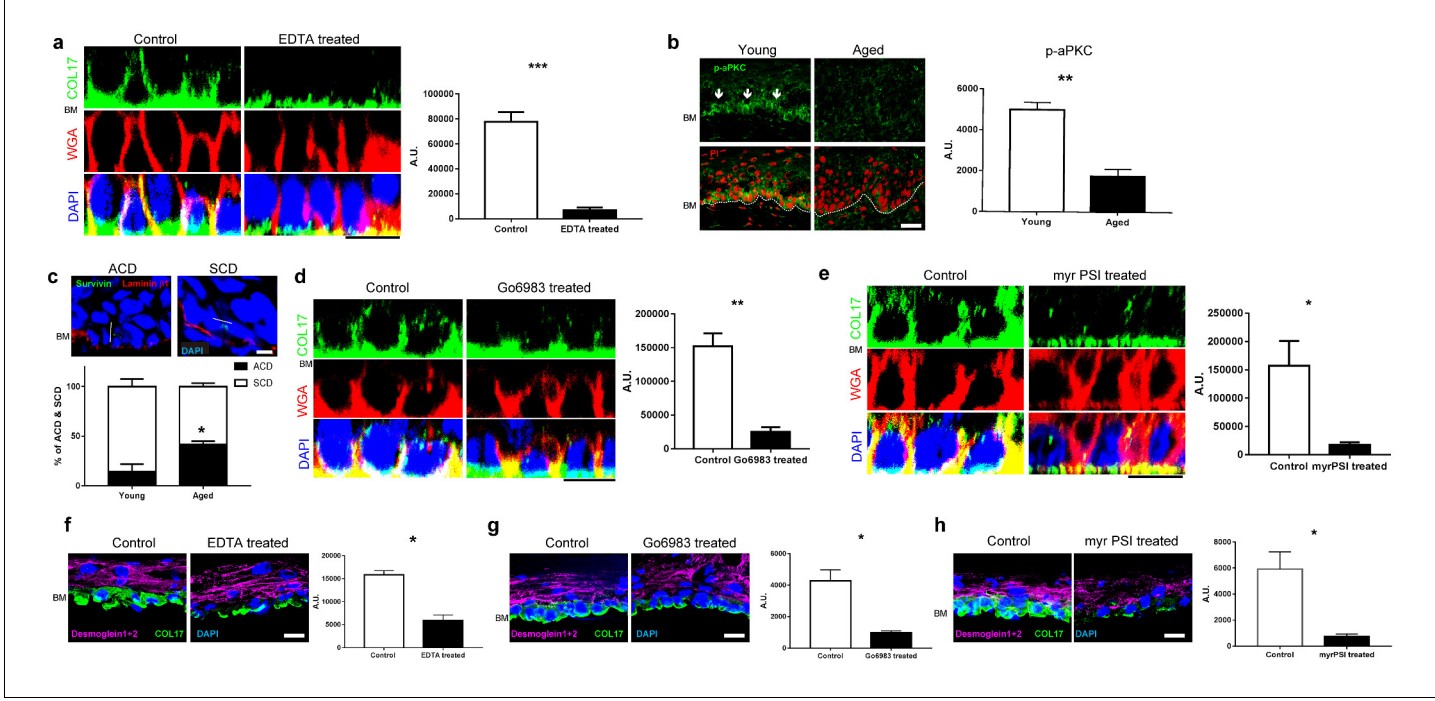

**Figure 5.** COL17 distribution is modulated by aPKC. (a) COL17 staining of whole IFE skin treated with 5 mM EDTA. The quantitative fluorescent intensity of COL17 in basal cells of IFE from control (PBS) and 5 mM EDTA treated (n = 4). Mann-Whitney test. Samples were taken from young adult WT mice at 6–10 weeks. Scale bar: 20 μm. (b) Phospho-aPKC labeling (indicated with arrows) and quantitative fluorescent intensity results from young and aged WT IFE skin (n = 4). Mann-Whitney test. Scale bar: 20 μm.(c) Representative figures of asymmetric cell division (ACD; scored as perpendicular to basement membrane) and symmetric cell division (SCD; in parallel to basement membrane) in young IFE. Survivin staining indicates the direction of the cell division. Laminin β1 signifies basement membrane. Scale bar: 10 μm. Graph of percentage of ACD and SCD in young and aged IFE (n = 4). Student's t-test. (d–e) The pharmacological inhibition of pan-aPKC (d, 1 μM of Go6983; 0.00002% DMSO as control) and aPKCλ/ζ (e, 10 μM of myr PSI; water as control) in whole IFE skin from young adult WT mice, followed by COL17 staining. The quantitative fluorescent intensity of COL17 in lateral membrane of basal cells from control and 1 μM Go6983 treated (d) and 1 μM myr PSI (e) (n = 4). Mann-Whitney test. Scale bar: 10 μm. BM, basement membrane. (f–h) EDTA treatment (5 mM; PBS as control, f) and pharmacological inhibition of pan-aPKC (g, 1 μM of Go6983; 0.00002% DMSO as control) and aPKCλ/ζ (h, 10 μM of myr PSI; water as control) on 3D epidermis. The relative fluorescent intensity of COL17 in lateral membrane of basal cells was measured (n = 4). Mann-Whitney test. BM, basement membrane. Scale bar: 20 μm. The data are the means ± SE. *0.01<p<0.05, **0.001<p<0.01, ***0.0001<p<0.001.

aPKC, an active form of aPKC, was reduced in IFE with aging (*Figure 5b*). Consistent with this finding, the axis of cell division in the aged IFE tended toward asymmetric cell division (ACD) (*Figure 5c*). This phenotype emulates the adult IFE of aPKCλ conditional-null mice (*Niessen et al., 2013*). To recapitulate aged skin with attenuated aPKC, we treated young mouse IFE samples with Go6983, a pan-PKC inhibitor, and myristoylated pseudosubstrate inhibitor (myr PSI), which is specific for aPKCζ and aPKCλ/ι (*Gschwendt et al., 1996*; *Helfrich et al., 2007*; *Standaert et al., 1999*). According to the whole mount staining, the pharmacological inhibition of aPKC diminished the apico-lateral COL17 in basal cells (*Figure 5d–e*). EDTA treatment and aPKC inhibition also diminished apico-lateral membranous COL17 in basal cells of reconstructed human 3D epidermis (*Figure 5f–h*). These data indicate that an aging-induced alteration of aPKC contributes to spatial changes in COL17 in the IFE.

COL17 undergoes posttranslational modifications, such as ectodomain shedding and degradation, in physiological and pathological settings through the action of several proteases, such as disintegrin and metalloproteinases 9, 10, and 17 (ADAM9/10/17), matrix metalloproteinase-9 (MMP9), neutrophil elastase (ELANE) and other serine proteases (*Hirako et al., 1998*; *Nishie, 2014*). There were no discernible changes between aged and young IFE skin in the expression levels of the proteases genes known to degrade COL17 (*Figure 4—figure supplement 4a*). The involvement of MMP9 in COL17 degradation in aged IFE was further excluded by the observation of the IFE in

Serpine1−/− mice, which lack plasminogen activator inhibitor-1, blocking MMP9 activity (*Figure 4—figure supplement 4b*). ELANE, which degrades COL17 adjacent to HFSCs (*Matsumura et al., 2016*), was located along the BMZ of the IFE, an expression pattern different from that is seen in hair follicles and back skin (*Figure 4—figure supplement 4c*). Physical aging did not affect ELANE labeling in the IFE. These data suggest that proteases may not play major roles in the loss of apico-lateral COL17 in the IFE.

## Overexpression of human COL17 ameliorates hyperproliferative state in the aged epidermis

Next, we tested whether hCOL17 overexpression could reverse the aged IFE phenotype. K14-hCOL17 transgenic mice, which overexpress hCOL17 under the K14 promoter, showed COL17 on all surfaces of IFE basal cells at an age >19 months (*Figure 6a*). In association with this observation, the epidermis was thinner, and the numbers of PH3- and PCNA-positive cells were reduced in the K14-hCOL17 aged IFE compared to the wild type (*Figure 6b*). However, the expression levels of *Itga6* and *Itgb1* were similar in K14-hCOL17 aged IFE and age-matched controls (*Figure 6c*). These data suggest that COL17 might act downstream of these stem cell markers and can keep the IFE in a juvenile state when overexpressed. Premature aged *Col17a1−/−* IFE (3 months old, *Figure 4—figure supplement 1*) exhibited epidermal thickening with an increased number of BrdU-positive basal cells, while the numbers of PCNA- or PH3-positive cells were comparable with age-matched controls (*Figure 6d*). This result was akin to wild-type aged IFE with a loss of apico-lateral COL17 (*Figure 4c*). Unlike neonatal IFE hypertrophy via inactive Wnt signaling, there were no obvious alterations in Wnt-related molecules in the young adolescent IFE (*Figure 6—figure supplement 1*). These results demonstrate that COL17 overexpression suppresses epidermal hyperproliferation associated with physical aging in the IFE.

## Discussion

BMZ proteins, including COL17, are key attachment structures that support epidermal cell homeostasis throughout life (*Watt and Fujiwara, 2011*). Previously, it has been suggested that COL17 serves as a niche for HFSCs (*Matsumura et al., 2016*), which subsequently maintain melanocyte stem cells (*Tanimura et al., 2011*). Here, we present the unidentified role of COL17 in regulating homeostasis in the paw IFE in terms of proliferation (*Figure 7*). COL17 deletion in the neonatal epidermis led to the transient hyperproliferation of the IFE, mediated via waned Wnt-$\beta$-catenin signaling. Physical aging led to increased IFE proliferation and the loss of the non-hemidesmosomal COL17 distribution associated with altered aPKC signaling. The restoration of COL17 in aged IFE reversed the hyperproliferative state.

Wnt-$\beta$-catenin signaling is one of master regulators of skin development and homeostasis in several components, including hair follicles, other appendages and the IFE (*Lim and Nusse, 2013*; *Lu and Fuchs, 2014*). Activating and inactivating mutations of Wnt-$\beta$-catenin signaling molecules give rise to distinct phenotypes in the IFE of murine back skin in terms of proliferation and differentiation (*Lim and Nusse, 2013*). Recently, the relationship between the IFE of murine paw skin and Wnt-$\beta$-catenin signaling was described (*Lim et al., 2013*). IFE basal cells require Wnt-$\beta$-catenin signaling to proliferate and express both Wnt ligands and inhibitors by an autocrine process in adolescent mice (*Lim et al., 2013*). Our study uncovers the role of COL17-stabilized Wnt signaling in governing neonatal IFE proliferation, although the interaction between COL17 and Wnt-related molecules remains elusive. Our observation of IFE hyperproliferation in the neonatal paw under inactive Wnt signaling (either by genetically modified mice or by inhibitors) is in contrast to the observed IFE hypotrophy in 1-month-old mice with inducible $\beta$-catenin loss-of-function (*Lim et al., 2013*). This discrepancy might be due to the difference of the observational time points (neonate vs 1-month-old) and/or of the model systems used in each study (conventional vs inducible).

Integrins, which are also BMZ components, have been implicated as having functional roles in regulating epidermal proliferation and differentiation (*Margadant et al., 2010*; *Watt, 2014*). Interestingly, mice and human epidermis deficient in Kindlin-1, an integrin co-activator, also show altered Wnt and TGF-$\beta$ signaling (*Rognoni et al., 2014*). In contrast to *Col17a1−/−* IFE, the absence of Kindlin-1 resulted in the activation of Wnt signaling in the IFE. The previous and current findings

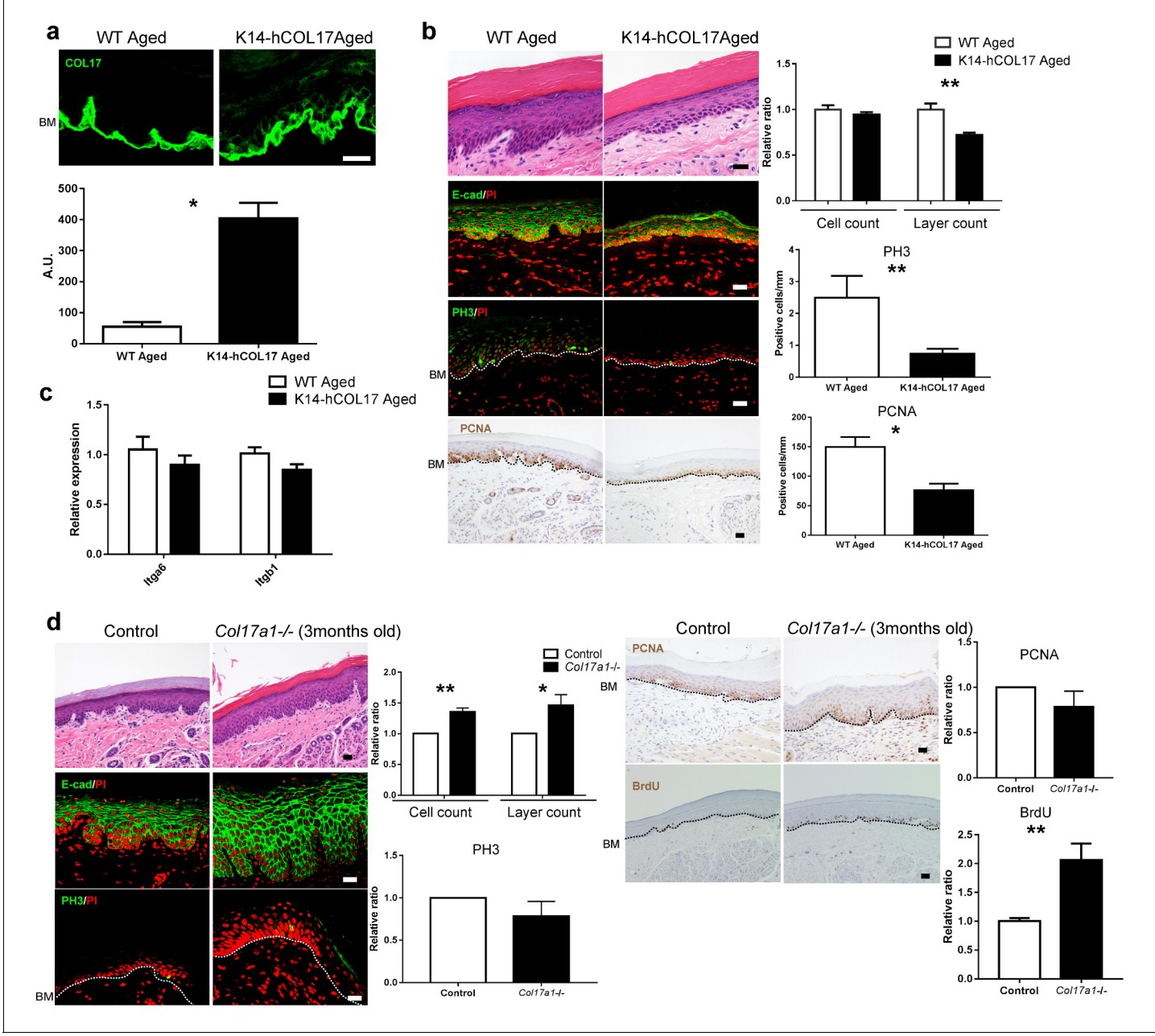

**Figure 6.** Overexpression of human COL17 ablates hyperproliferation in aged IFE. (a) COL17 labeling and its quantitative fluorescent intensity in IFE skin from WT and K14-hCOL17 aged mice (>19 months old) (n = 5). The antibody used in this assay detects both human and murine COL17. Scale bar: 20 μm. Mann-Whitney test. (b) H&E-, E-cad-, PH3- and PCNA-stained skin samples from WT and K14-hCOL17 aged IFE. The numbers of epidermal layers, total epidermal cell counts, and PH3- and PCNA-positive basal cell counts (n = 5 for H&E, E-cad, and PCNA staining; n = 4 for PH3 in aged K14-hCOL17 mice; n = 3 for PH3 of aged WT mice). Student's t-test. Scale bar: 20 μm. (c) The gene expression levels of *Itga6* and *Itgb1* in IFE skin samples from WT and K14-hCOL17 aged IFE (n = 5). Student's t-test. (d) H&E-, E-cad-, PH3- and PCNA-staining; quantifications of epidermal layers; total epidermal cells; and PH3-, PCNA-, and BrdU-positive basal cells in the IFE of *Col17a1−/−* mice and littermate controls (3 months old) (n = 4). Student's t-test. Scale bar: 20 μm. The data are the means ± SE. *0.01<p<0.05, **0.001<p<0.01.

The following figure supplement is available for figure 6:

**Figure supplement 1.** qRT-PCR assessment of Wnt-related molecules in IFE skin samples from *Col17a1−/−* and littermates at the age of 12 weeks.

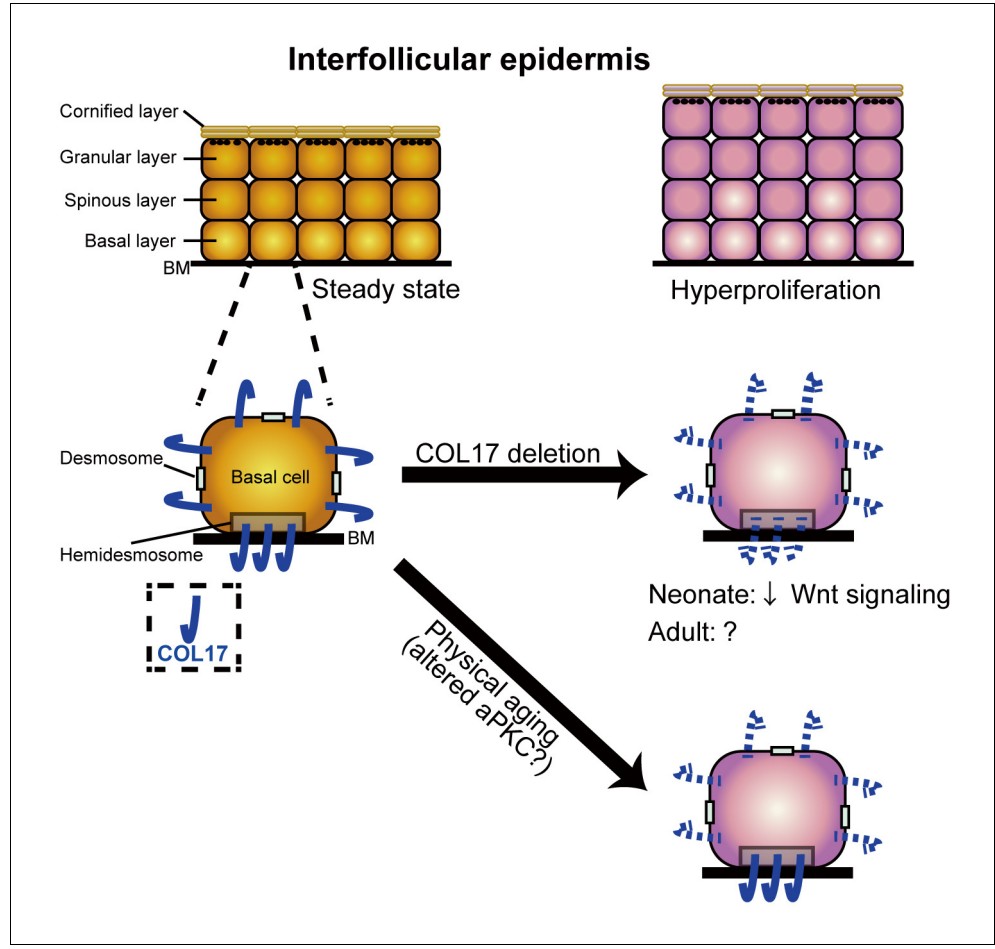

**Figure 7.** A model of the role of COL17 in maintaining IFE homeostasis. A graphical abstract of this study. COL17 regulates paw IFE homeostasis in coordination with Wnt signaling at the neonatal stage. Physical aging diminishes non-hemidesmosomal COL17 labeling in IFE keratinocytes, leading to IFE hyperproliferation, associated with altered aPKC activities.

suggest that the regulation of Wnt signaling in the IFE is subject to complex regulation by the epidermal BMZ.

For epidermal maintenance and proliferation, the presence of epidermal stem cells is essential. However, the theories on how these stem cells reside in the epidermis and how they regulate epidermal homeostasis remain controversial and may be dependent on the location on the body (*Mascré et al., 2012*; *Rompolas et al., 2016*; *Sada et al., 2016*; *Sánchez-Danés et al., 2016*). In the IFE, it is proposed that the majority of basal cells exist as plausible committed progenitors (CPs), which undergo cell division and differentiation, in part through asymmetric cell divisions (*Alcolea and Jones, 2014*; *Doupé and Jones, 2012*). In the foot pad, the proliferative potentials of CPs are elevated compared to other epidermal sites, and the clones of foot pad CPs are also enlarged in aged skin (*Dunnwald et al., 2003*, *2001*; *Stern and Bickenbach, 2007*). Other in vitro studies report the expansion of proliferative CPs in the aged epidermis (*Charruyer et al., 2009*). Our data on hypertrophic IFE in aged skin reinforce the notion that physical aging abolishes the quiescent state of the IFE. This hypothesis requires clarification using conventional fluorescent lineage tracking and/or genetics-based cell fate mapping (e.g. sequencing lineage barcodes) (*Woodworth et al., 2017*). Moreover, in our study, the introduction of hCOL17 into aged IFE ablates epidermal hyperproliferation, which is in line with the rejuvenation of HFSCs upon hCOL17 overexpression (*Matsumura et al., 2016*). These findings highlight the role of COL17 as an anti-aging molecule in the skin.

As the epidermis is a highly structured tissue, epithelial polarity is required for homeostasis (*Tellkamp et al., 2014*). Although physiological aging is reported to affect aPKCs in neurons (*Pascale et al., 2007*), the alteration of aPKC in the epidermis upon aging is poorly understood. Our study adds skin to the list of organs in which aPKC is modified with aging. It is noteworthy that in early adult aPKCλ-deficient mice, the IFE showed a similar proliferative phenotype to those of aged WT IFE (*Niessen et al., 2013*; *Osada et al., 2015*). Asymmetric cell divisions, rather than symmetric cell division, are increased in these mice and are thought to force the gradual loss of quiescence, contributing to premature aged phenotypes (*Tellkamp et al., 2014*). Altered COL17 distribution with aging or with pharmacological inhibition of aPKC might result from improper cell polarity. To further confirm our results, the observation of COL17 in an aPKCλ-deficient epidermis would be critical as we cannot completely exclude the involvement of non-specific drug effects on the epidermis in COL17 dynamics.

Recently, aging has been illuminated as a trigger for the clonal expansion of stem cells, which disturbs tissue health by means of oncogenesis (*Goodell and Rando, 2015*). The accumulation of Ki-67-positive cells and prolonged hyperplasia in the aged epidermis during recovery from carcinogenic treatments has been reported (*Golomb et al., 2015*). It would be intriguing to observe skin carcinogenesis upon COL17 deficiency, as its relevance in colon and lung cancers has been described (*Liu et al., 2016a*, *2016b*; *Moilanen et al., 2015*).

To further elucidate the mechanisms and dynamics of COL17 in regulating IFE homeostasis, an investigation with either drug-inducible expression or loss of COL17 expression under specific promoters is warranted. The discrepancy in the epidermal growth rate with aging between hyperproliferation, as observed in our findings and the reports of others (*Charruyer et al., 2009*; *Stern and Bickenbach, 2007*), and hypoproliferation (*Giangreco et al., 2008*) might be attributed to the following: (1) UV exposure, which inhibits cell growth, on human skin; (2) a wide divergence of epidermal thickness among the different locations on the body (*Porter et al., 1998*) and among individual people (*Waller and Maibach, 2005*); (3) differences in hygiene status among the animal facilities and (4) the influence of the development and cyclical growth of hair on the IFE of haired skin. Our observations can exclude the involvement of most of these extrinsic and intrinsic confounding factors by restricting the monitoring to the paw IFE of congenic mice at a single animal facility. However, it is noteworthy that chronic stimulation via ambulation might affect paw IFE homeostasis with physical aging, which is a limitation of our study. Considering chronic stimulation and lack of UV exposure, it would be fair to say that our data on the murine paw IFE could be extrapolated into the human palmoplantar and buttock epidermis.

In closing, our study has revealed an unrecognized link between COL17 and epidermal proliferation in neonatal and aged IFE. We propose that COL17 is a good candidate to target to prevent epidermal aging and oncogenesis.

## Material and methods

### Animals
*Col17a1−/−* (RRID:MGI:3711939), K14-hCOL17 (a courtesy gift of Prof. Kim B Yancey), hCOL17+; *Col17a1−/−* (RRID:MGI:3711948), and K14-deltaNLef1 (RRID:MGI:2667413) mice were generated as previously described (*Niemann et al., 2002*; *Nishie et al., 2007*). ins-TOPGAL+ mice (RRID:IMSR_RBRC05918) (*Moriyama et al., 2007*) were obtained from the RIKEN BRC (Tsukuba, Ibaraki, Japan) and bred with *Col17a1−/−* mice to produce ins-Topgal+;*Col17a1−/−* mice. C57BL/6 strain mice and *Klotho−/−* (RRID:MGI:2181617) mice were purchased from Clea (Tokyo, Japan). *Serpine1−/−* mice (RRID:IMSR_JAX:002507) were purchased from The Jackson Laboratory (Bar Harbor, Maine, USA).

### BrdU labeling for proliferation analysis
For label analysis, 1-day-old mice were intraperitoneally administered 10 µg BrdU (BD Pharmingen, New Jersey, USA) per head, and adult mice were intraperitoneally administered 8.33 µg/g per head 4 hr before sacrifice.

## Antibodies

The following antibodies were used: anti-E-cadherin (Cell Signaling Technology, Danvers, Massachusetts, USA; 24E10, RRID: AB_10694492), anti-phospho histone H3 (Ser10) (Merck Millipore, Billerica, Massachusetts, USA, RRID: AB_11210699), anti-PCNA (Dako, Santa Clara, California, USA; PC10, RRID: AB_2160651), anti-BrdU (Dako; M0744, RRID:AB_10013660), anti-loricrin (Covance, Princeton, New Jersey, USA, RRID:AB_10064155), anti-cleaved caspase-3 (Cell Signaling Technology, RRID:AB_2341188), FITC-conjugated anti-CD3e (BioLegend, San Diego, California; 145–2 C11, RRID:AB_394595), Alexa Fluor 488-conjugated anti-F4/80 (Affymetrix, Santa Clara, California, USA; BM8, RRID: AB_893479), FITC (fluorescein isothiocyanate)-conjugated anti-Ly-6G (Beckman coulter, Brea, California, USA; RB6-8C5,RRID: AB_394643), anti-TGF$\beta$1 (Santa Cruz Biotechnology, Dallas, Texas, USA, RRID: AB_632486), anti-p-Smad2 (Cell Signaling Technology; 138D4, RRID:AB_490941), anti-LEF1 (Cell Signaling Technology; C12A5, RRID: AB_823558), anti-$\beta$-catenin (BD; 14/Beta-catenin, RRID: AB_397554), anti-hCOL17 NC16A domain (TS39-3, homemade) (*Ujiie et al., 2014*), anti-extracellular portion of hCOL17 (mAb233, homemade) (*Nishizawa et al., 1993*), anti-murine COL17 NC14A domain (MoNC14A, homemade) (*Nishie et al., 2015*), anti-cytoplasmic portion of COL17 (Abcam, Cambridge, UK; ab186415, 1A8c, homemade (*Nishizawa et al., 1993*)), anti-plectin (Abcam; ab32528, RRID: AB_777339), anti-BP230 (Cosmo bio, Tokyo, Japan; 239, RRID:AB_1961833), anti-desmogleins 1 and 2 (PROGEN, Wieblingen, Heidelberg, Germany; DG3.10, RRID: AB_1284107), anti-integrin $\beta$1 (Chemicon International, Billerica, Massachusetts, USA; MB1.2, RRID: AB_2128202), anti-integrin $\alpha$6 (BD Pharmingen; GoH3, RRID: AB_2296273), anti-ELANE (Abcam), anti-phospho aPKC (Santa Cruz; sc-271962, RRID:AB_10708397), and anti-survivin (Cell Signaling Technology; 71G4B7, RRID:AB_10691694), anti-laminin $\beta$1 (Abcam; LT3, RRID: AB_775971).

## Histology

Specimens from mice paw skin and human skin were fixed in formalin and embedded in paraffin after dehydration or were frozen on dry ice in an optimal cutting temperature (OCT) compound. Frozen sections were fixed with 4% paraformaldehyde (PFA) or acetone or were stained without fixation. Antigen-retrieval with pH 6.0 (citrate) or pH 9.0 (EDTA) buffer was performed on deparaffinized sections. Sections were incubated with the primary antibodies overnight at 4°C. After washing in phosphate-buffered saline (PBS), the sections were incubated with secondary antibodies conjugated with Alexa488, Alexa647 or FITC for 1 hr at room temperature (RT). The nucleus was stained with propidium iodide (PI) or 4′,6-diamidino-2-phenylindole (DAPI). All the stained immunofluorescent samples were observed using a confocal laser scanning microscope (Olympus Fluoview FV1000; Olympus, Tokyo, Japan, RRID:SCR_014215). To quantify the relative intensity of the staining, the signals for immunofluorescence were obtained and analyzed using FV1000 software (RRID:SCR_014215) or Image J (NIH, Bethesda, Maryland, USA, RRID:SCR_003070). The values were normalized by the section length or by the tissue area and are indicated as arbitrary units (A. U.).

For immunohistochemistry, horseradish peroxidase (HRP)-tagged antibodies were used. Sections were blocked before antibody labeling and counterstained with hematoxylin. Images were captured with a bright field microscope (Nikon, Tokyo, Japan; Keyence, Tokyo, Japan).

For staining, skin samples of normal humans, both young (<10 years old) and aged (>85 years old), were obtained from non-sun-exposed areas. To distinguish hemidesmosomal from non-hemidesmosomal COL17 in basal keratinocytes, frozen skin sections were treated with 0.5% Triton X-100 in PBS for 1 hr at RT before primary antibody incubation, as described previously (*Hirako et al., 1998*).

To evaluate the skin thickness, at least three footpad areas per each mouse were counted and averaged. Epidermal cells were quantified as the number of PI-positive cells per epidermal length. The number of epidermal cell layers was evaluated using E-cadherin labeling. The numbers of basal cells positive for proliferation markers and Wnt-related molecules were counted and normalized using the total epidermal length or total basal cell number. For general histological analysis, each immunostaining was repeated in at least three independent experiments, and sweat gland areas were excluded from observation.

For high-resolution structured illumination microscopy (SIM) imaging, an N-SIM microscope (Nikon) with an electron-multiplying charged-coupled device camera (DU-897; Andor Technology,

Tokyo, Japan) was used as described previously (*Hashimoto et al., 2016*). Image reconstruction was performed using NIS-Elements software (RRID:SCR_014329).

In division axis orientation determination, the direction of cell division was verified using survivin staining as described previously (*Niessen et al., 2013*). The angle of division was confirmed by scaling the angle of the plane transecting two daughter cells relative to the plane of the basement membrane as labeled laminin $\beta$1. The total number of cell divisions including ACD and symmetrical cell division (SCD) were set to 100% per sample.

## TOPFLASH assay

HEK293 Cells (RRID:CVCL_0045, authenticated by STR gene profiling) which constitutively expresses the SuperTopFlash construct (SuperTopFlash 293 (STF293)) (*Tsukiyama et al., 2015*) was cultured in Dulbecco's Modified Eagle medium (DMEM) supplemented with 10% bovine serum (Sigma-Aldrich, St. Louis, Missouri, USA) and 1% penicillin-streptomycin-amphotericin B (Wako, Osaka, Japan). The cells used in this assay were subject to regular mycoplasma testing. Wnt 3a-conditioned medium (Wnt3a CM) was kindly provided by S. Takada. Full-length human *COL17A1* cDNA (NM_0004940) was subcloned into the mammalian expression vector pcDNA3.1/Zeo (ThermoFisher, Waltham, Massachusetts, USA) and transfected into STF293 cells. The transfected cells were selected with Zeocin (Thermo Fischer Scientific, Waltham, Massachusetts, USA). STF293 cells stably expressing either human *COL17A1* or empty vector were seeded onto 96-well plates and co-transfected with the pGL 4.75 plasmid, which contains a Renilla reniformis luciferase gene (Promega, Fitchburg, Wisconsin, USA), using Lipofectamine2000 (Invitrogen, Waltham, Massachusetts, USA) and Opti-MEM (Thermo Fischer Scientific). At 24 hr after seeding, Wnt3a CM was added to the medium at at a final concentration of 30% of the total medium by volume. After 2 days of stimulation with the Wnt3a CM, the Firefly and Renilla luciferase activities were measured using a Dual-Luciferase reporter assay system (Promega) and a Spectra Max Paradigm (Molecular device, Tokyo, Japan). All the values were normalized to Renilla luciferase activity.

## Wnt inhibitor treatment on WT neonate mice

Either IWP-2 (Sigma Aldrich) or Wnt-C59 (Cayman Chemical, Ann Arbor, Michigan, USA; both dissolved in DMSO) were intraperitoneally injected into WT mice at P0 (IWP-2: 10 mg/g body, Wnt-C59: 20 mg/g body) (*Carotenuto et al., 2017*; *Kuo et al., 2016*). The paw skin samples were collected 24 hr after injection.

## Whole mount staining

For X-gal staining on ins-Topgal+ mice skin, a beta-galactosidase staining kit (Takara-bio, Shiga, Japan) was used according to the provider's protocol. Briefly, hindpaw samples were fixed with 4% PFA for 2 hr at RT and soaked in staining solution overnight. Tissues were mounted with a Mowiol solution. Images for evaluation were collected from the heel of the hind paw to exclude the involvement of hair follicles and sweat glands. The individual light settings of each experiment were identical for both control and *Col17a1−/−* mice, when capturing images with a bright field microscopy (Nikon). The LacZ-positive area was quantified by ImageJ.

For epidermal whole mount staining, epidermal sheets were taken from whole paw mice skin as described previously with some modifications (*Kubo et al., 2009*). For cell membrane detection, Alexa633-conjugated wheat germ agglutinin (Invitrogen, Waltham, Massachusetts, USA) was used. Three-dimensional (3D) reconstruction images were generated, and analyses were performed using the Metamorph software (Molecular Devices, Tokyo, Japan, RRID:SCR_002368).

## Ex vivo inhibitor treatment

Whole mice paw skin samples were treated with the following inhibitors for 1 hr at 4°C: EDTA, pan-PKC inhibitor Go6983 (Tocris Bio, Bristol, UK) and myristoylated pseudosubstrate Inhibitor (myr PSI; Calbiochem, Billerica, Massachusetts, USA) before staining. These inhibitors were used at the indicated concentrations as previously described (*Amagai et al., 1995*; *Atwood et al., 2013*; *Wu et al., 2006*).

## Reconstituted 3D human epidermis inhibitor treatment

Reconstituted 3D epidermis (LabCyte EPI-MODEL) derived from cultured human epidermal keratinocytes was purchased from J-TEC (Aichi, Japan). Prior to pharmacological treatment, the reconstituted epidermis was cultured in the medium for 12 hr according to the manufacturer's protocol. The treatment details were as described in ex-vivo inhibitor treatment.

## Quantitative RT-PCR (qRT-PCR)

RNA was extracted from whole paw skin using the RNeasy Mini kit (QIAGEN), and the cDNA was prepared using the SuperScript III First-Strand Synthesis System (Thermo Fisher Scientific). The qRT-PCR was performed using the designated primers and fast SYBR Green (Thermo Fisher Scientific) in a STEP-One Plus sequence detection system (Applied Biosystems, Waltham, Massachusetts, USA). All the primers used in this study are listed in *Table 1*.

## Barrier function assay

An in situ skin permeability assay using Toluidine blue was performed as previously described (*Hardman et al., 1998*). Transepidermal water loss (TEWL) of the embryo back skin was measured using an evaporimeter (AS-VT100RS;Asahi Biomed, Tokyo, Japan) (*Yanagi et al., 2008*).

## Transmission electron microscopy

Biopsy samples from paw skin specimens were fixed in 2% glutaraldehyde solution, post-fixed in 1% $OsO_4$, dehydrated, and embedded in Epon 812. The embedded samples were sectioned at 1 μm

**Table 1.** Primers used in qRT-PCR.

| Gene name | Forward (5′→3′) | Reverse (5′→3′) |
|---|---|---|
| mTcf7l1 | TGGTCAACGAATCGGAGAAT | TCACTTCGGCGAAATAGTCG |
| mTcf7l2 | CTCCACAGCTCAAAGCATCA | CACCACCTTCGCTCTCATCT |
| mLef1 | CGCTAAAGGAGAGTGCAGCTA | GCTGTCTCTCTTTCCGTGCT |
| mTcf7 | GCCAGAAGCAAGGAGTTCAC | ACTGGGCCAGCTCACAGTAT |
| mCtnnb1 | AAGGCTTTTCCCAGTCCTTC | CCCTCATCTAGCGTCTCAGG |
| mFzd7 | GACCAAGCCATTCCTCCGTG | CAGGTAGGGAGCAGTAGGGTA |
| mFzd4 | AACCTCGGCTACAACGTGAC | GGCACATAAACCGAACAAAGGAA |
| mNfatc4 | CAAGCTGCGAGGATGAGGAG | ACAGCATGGAGGGGTATCCT |
| mWnt4 | GTCAGGATGCTCGGACAACAT | CACGTCTTTACCTCGCAGGA |
| mWnt5a | GGACCACATGCAGTACATTGG | CGTCTCTCGGCTGCCTATTT |
| mFzd3 | TGATGAGCCATATCCCCGACT | GCCTATGAAATAGCGAGCAAATG |
| mFzd8 | CCGCTGGTGGAGATACAGTG | CGGTTGTAGTCCATGCACAG |
| mItga6 | ATGCCACCTATCACAAGGCT | GCATGGTATCGGGGAATGCT |
| mItgb1 | ATCATGCAGGTTGCGGTTTG | TGGAAAACACCAGCAGTCGT |
| mTgm1 | TTTGATGGGTGGCAGGTTGT | GCCATTCTTGACGGACTCCA |
| mCol17 | GATGGCACTGAAGTCACCGA | TATCCATTGCTGGTGCTCCC |
| mPpl | GCATGCTGAGTGGAAGGAGT | AAGTCTGAGTCCACCTTGCG |
| mEvpl | TCCTACAAGCTGCAAGCACA | TCTAAGGAGCAGCGGTAGGT |
| mIvl | CTCCTGTGAGTTTGTTTGGTCT | CACACAGTCTTGAGAGGTCCC |
| mCyc1 | ATCGTTCGAGCTAGGCATGG | GCCGGGAAAGTAAGGGTTGA |
| mGusb | CAGGGTTTCGAGCAGCAATG | ACCCAGCCAATAAAGTCCCG |
| mGapdh | TCCTGCACCACCAACTGCTTAGC | TGGATGCAGGGATGATGTTCTGG |
| hCOL17 | TCAACCAGAGGACGGAGTCA | TCGACTCCCCTTGAGCAAAC |
| h18S | GGCGCCCCCTCGATGCTCTTAG | GCTCGGGCCTGCTTTGAACACTCT |

thickness for light microscopy and thin-sectioned for electron microscopy (70 nm thick). The thin sections were stained with uranyl acetate and lead citrate and examined by transmission electron microscopy (H-7100; Hitachi, Tokyo, Japan).

### *COL17A1* knockdown and cell proliferation assay on normal human epidermal keratinocytes

Normal human epidermal keratinocytes (NHEKs, Lonza) in passages 3–4 were seeded with KGM-Gold (Lonza) in a 96-well plastic bottom plate. The cells were transfected with 10 µM of either human *COL17A1* siRNAs or the control (Mock) (Silencer Select siRNAs, Thermo Fisher, Waltham, Massachusetts, USA) using RNAimax (Thermo Fisher) and Opti-MEM (Thermo Fisher). Cell growth curves were generated using an xCELLigence system (ACEA bioscience, San Diego, California, USA, RRID:SCR_014821) at 24 hr after the knockdown procedure.

For the colony formation assay, 1000 cells/ml of NHEKs (passages 3–4) were seeded in 6-well plates on mitomycin C (Wako, Osaka, Japan) treated 3T3-J2 feeder cells (RRID:CVCL_W667) and were transfected with either *COL17A1* siRNA or mock siRNA, as described above. Cells were maintained in KGM-Gold with 10% fetal bovine serum (Sigma-Aldrich, St. Louis, Missouri, USA) for 2 weeks. To detect the colonies, cells were fixed with 4% paraformaldehyde for 10 min and stained for 20 min with 1% rhodamine (Sigma-Aldrich) diluted in distilled water. The colony number and size were analyzed by ImageJ (NIH, Bethesda, Maryland, USA). All the cells used in this assay were subject to regular mycoplasma testing.

### Immunoblotting

Cultured cells were collected and lysed in 1% NP-40-containing buffer. The samples were loaded on a NuPAGE 4–12% Bis-Tris gel (Thermo Fisher) and then transferred to a PVDF membrane. The membrane was incubated with primary antibodies (anti-C-terminal region of human COL17 (09040) (*Ujiie et al., 2014*) and anti-beta tubulin (Abcam, Cambridge, UK, RRID:AB_2210370)) followed by secondary antibodies conjugated with horseradish peroxidase. The blots were detected using ECL-Plus (GE Healthcare).

### Subjects with JEB

Skin samples from two patients with the generalized other subtype of JEB were used in this study. The JEB patients harbored truncation mutations in COL17A1, and their skin specimens were negative for COL17 expression (*Nakamura et al., 2006*)(Masuda et al., manuscript in preparation). The patients were a newborn Japanese male and a 15-year-old Japanese male. As controls for the immunofluorescence staining, skin samples were collected from age-, site-, and sex-matched healthy individuals.

### Gene expression microarray

The total RNA extracted from young (6- to 10-week-old) and aged (19- to 27-month-old) mouse whole paw skin was hybridized to an Agilent SurePrint G3 Mouse v2 8 × 60K 1 color 8 microarray. N = 4 for each group. The microarray data were analyzed using GeneSpring software (Agilent Technology, Santa Clara, California, USA, RRID:SCR_009196).

### Data availability

Microarray data are available in the ArrayExpress database (www.ebi.ac.uk/arrayexpress, RRID:SCR_002964) under accession number E-MTAB-4916.

### In silico model

The authors Y. K. and M. N. previously introduced a mathematical model that could simulate the homeostasis of the epidermis (*Kobayashi et al., 2016*). In this model, the four epidermal processes were taken into account: (i) the kinetic interactions of epidermal cells, (ii) the reproduction of stem cells and daughter cells, (iii) differentiation, and (iv) calcium dynamics. Migration of the cells after leaving the basal layer was naturally realized by the hard-core repulsive interactions among the cells, and the differentiation process was controlled by the calcium dynamics.

Numerical investigation of the model revealed that because of the calcium dynamics, the thickness of the epidermis was maintained, and the lower boundary of the stratum corneum retained a flat structure.

For the present purpose, we modified this model in such a way that we could investigate the effect of the binding strength of the stem cells to the basal membrane. The kinetic interactions and the reproduction process were modified as follows.

Each cell was represented as a sphere characterized by its position $x_i = (x_i,\ y_i,\ z_i)$ and radius $R$. The basal membrane was modeled as a monolayer of densely packed immobile particles, each having the radius $R_m$. The cells obeyed the following equations of motion:

$$\mu \frac{dx_i}{dt} = -\frac{\partial}{\partial x_i}\left( \sum_{j\in\Omega_{LJ}} V_{LJ}\left(|x_i - x_j|\right) + \sum_{j\in\Omega_m} V_m\left(|x_i - x_j|\right) \right),$$

where $\mu$ is the coefficient of friction; $V_{LJ}$ is the repulsive interaction between cell $i$ and cell $j$, which is either one of the adjacent cells or the nearest membrane particle; and $V_m$ is the interaction with the membrane. The interactions were considered only for the particles that are in contact with cell $i$. The first summation was taken over the set $\Omega_{LJ}$, which consists of both cells and membrane particles, and the second summation was over $\Omega_m$, which contains only the membrane particles.

The first potential was given by

$$V_{LJ}(r) = \varepsilon\left\{ \left(\frac{R+R_j}{r}\right)^6 - \frac{1}{2}\left(\frac{R+R_j}{r}\right)^{12} \right\},$$

where $\varepsilon$ is a positive constant. If cell $j$, interacting with $i$, represented the epidermal cell, then $R_j = R$, and if cell $j$ is the membrane particle, then $R_j = R_m$. The second potential $V_m$ was assumed to be cell-dependent. For the suprabasal cells, $V_m = 0$; for the stem cells, the interaction was spring-like:

$$V_m(r) = \frac{K_s}{2}\left\{(r - (R+R_m)\right\}^2,$$

where $K_s$ is a positive constant. For the daughter cells, it was assumed to be a nonlinear spring:

$$V_m(r) = \begin{cases} \frac{K_d}{2}\left[ a\{(r-(R+R_m)\}^2 - \frac{b}{2}\{(r-(R+R_m)\}^4 \right] & (r \le r^\star), \\ 0 & (r > r^\star), \end{cases}$$

where $K_d$, $a$, and $d$ are positive constants and $r^* = R + R_m + \sqrt{a/b}$. That is, the daughter cells could leave the basal layer when the distance became greater than the threshold $r^*$, while the stem cells were bound to the basal membrane and could not leave it.

The cell reproduction process was modeled in the following way: each reproducible cell was assigned a deterministic cell cycle with the period $T$. After the period elapsed, the cell entered into the stochastic division process, which was the Poisson process with the rate $\gamma$.

When the division of cell $i$ started, the new particles $j$ and $k$ were created with the radius $R$ at the same place as cell $i$, with cell $i$ itself deleted: $x_j(t_0) = x_k(t_0) = x_i(t_0)$. Note that the overlapping of $j$ and $k$ was allowed. The force exerted on $j$ by $k$ was given as the partial derivative $-\frac{\partial}{\partial x_j}$ of the following elastic potential:

$$U\left(|x_j - x_k|\right) = \frac{K'}{2}\left(|x_j - x_k| - l(t)\right)^2,$$

where $K'$ is a positive constant and $l(t)$ is the natural distance between $j$ and $k$, which linearly increases as $l(t) = \alpha(t - t_0)$ with a constant $\alpha$. When $l(t)$ reached $l(t) = 2R$, the division was considered to be completed, and the potential $U$ was no longer computed.

The differentiation processes and calcium dynamics were the same as in the manuscript (*Kobayashi et al., 2016*), which were summarized as follows: each suprabasal cell was assigned a degree of differentiation $S$ as an internal variable. When $S$ reached the threshold $S^*$, the cell underwent terminal differentiation and became a cornified cell. The advancement of $S$ was accelerated

when the calcium concentration of the cell was high, and calcium excitation was induced when the cell was in contact with the cornified cells, reflecting the fact that high-calcium concentration was observed beneath the stratum corneum.

We numerically solved this model with the following initial conditions. According to previous results, we first obtained the steady states of the epidermis with an undulated basal membrane. We used these states as initial conditions. For each of five different initial conditions, we decreased the binding strength, $K_d$, to the basal membrane from $K_d = 5.0$ to $K_d = 2.0$ and ran simulations. For comparison, we also ran simulations for each of the initial conditions without changing $K_d = 5.0$. Other parameters were chosen: $\mu = 1.0$, $R = 1.4$, $R_m = 1.0$, $\varepsilon = 1.0$, $K_s = 25.0$, $a = 0.0868$, $b = 0.376$, $K' = 5.0$, $\gamma = 0.00813$, and $\alpha = 0.14$. By choosing these parameters and others used in the previous work [1], we assumed that the average cell division and turnover periods were approximately 3 and 28 days, respectively.

*Figure 1F* shows the ratio of the epidermis to its initial value. Each curve represents the ensemble average of five independent simulations with different shapes of the basal membrane.

## Statistical analysis

Statistical analysis was performed using GraphPad Prism (GraphPad Software, La Jolla, California, USA, RRID:SCR_002798). p-Values were determined using Student's t-test, the Mann-Whitney test or one-way ANOVA followed by Tukey's test. p-Values are indicated as *$0.01<p<0.05$, **$0.001<p<0.01$, ***$0.0001<p<0.001$, ****$p<0.0001$. The values are shown as the means ± standard errors (SE). Sample size for animal experiments was determined on the basis of pilot experiments.

## Acknowledgements

We thank Dr. Tsukasa Oikawa for his valuable support and insights. We thank Meari Yoshida for her technical assistance. We also thank Professor Kim B Yancey for providing K14-hCOL17 mice, Professor Yumiko Saga for providing the ins-Topgal+ mice and Professor Shinji Takada for Wnt3a-conditioned medium. The high-resolution SIM imaging was supported by Nikon. This work was funded by the Japan Foundation for Aging and Health, the Rohto Dermatology Research Award, JSID's Fellowship Shiseido Research Grant, the Cosmetology Research Foundation, the Geriatric Dermatology Research Grant, the Akiyama Foundation and JST CREST (JPMJCR15D2).

## Additional information

### Competing interests

FMW: Deputy editor, *eLife*. The other authors declare that no competing interests exist.

### Funding

| Funder | Grant reference number | Author |
| --- | --- | --- |
| Japanese Foundation for Aging and Health | | Ken Natsuga |
| Rohto Dermatology Research Award | | Ken Natsuga |
| Japanese Society for Investigative Dermatology | JSID's Fellowship Shiseido Research Grant | Ken Natsuga |
| Cosmetology Research Foundation | | Ken Natsuga |
| Geriatric Dermatology Research Grant | | Ken Natsuga |
| Akiyama Life Science Foundation | | Ken Natsuga |
| Japan Science and Technology Agency | CREST JPMJCR15D2 | Masaharu Nagayama |

The funders had no role in study design, data collection and interpretation, or the decision to submit the work for publication.

## Author contributions

MW, Conceptualization, Data curation, Formal analysis, Validation, Visualization, Writing—original draft, Writing—review and editing; KN, Conceptualization, Data curation, Formal analysis, Supervision, Funding acquisition, Validation, Investigation, Writing—original draft, Project administration, Writing—review and editing; WN, FMW, Resources, Supervision, Writing—review and editing; YK, Data curation, Writing—original draft; GD, Resources, Data curation, Writing—review and editing; SSu, Data curation, Formal analysis, Investigation, Visualization, Writing—review and editing; YF, Data curation, Investigation, Writing—review and editing; TT, MN, Resources, Formal analysis, Writing—review and editing; HU, Resources, Formal analysis, Investigation, Writing—review and editing; SSh, Investigation, Writing—review and editing; HN, Resources, Data curation, Visualization, Writing—review and editing; MM, MO, Resources, Writing—review and editing; HS, Supervision, Writing—original draft, Writing—review and editing

## Author ORCIDs

Ken Natsuga, http://orcid.org/0000-0003-3865-6366
Fiona M Watt, http://orcid.org/0000-0001-9151-5154

## Ethics

Human subjects: The institutional review board of the Hokkaido University Graduate School of Medicine approved all human studies described above. The study was conducted according to the Declaration of Helsinki Principles. Participants or their legal guardians gave their written informed consent.

Animal experimentation: The institutional review board of the Hokkaido University Graduate School of Medicine approved all animal studies described above.

# Additional files

## Major datasets

The following dataset was generated:

| Author(s) | Year | Dataset title | Dataset URL | Database, license, and accessibility information |
|---|---|---|---|---|
| Ken Natsuga | 2017 | Aging effect on interfollicular epidermis (paw skin) | https://www.ebi.ac.uk/arrayexpress/experiments/E-MTAB-4916/ | Publicly available at the EBI European Nucleotide Archive (accession no: E-MTAB-4916) |

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
