## [Decision Letter]

[Editors’ note: a previous version of this study was rejected after peer review, but the authors submitted for reconsideration. The first decision letter after peer review is shown below.]

Thank you for submitting your work entitled "Type XVII collagen coordinates proliferation in the interfollicular epidermis" for consideration by *eLife*. Your article has been reviewed by three peer reviewers, and the evaluation has been overseen by Reviewing Editor Johanna Ivaska and a Senior Editor. The reviewers have opted to remain anonymous.

Our decision has been reached after consultation between the reviewers. Based on these discussions and the individual reviews below, we regret to inform you that your work will not be considered for publication in *eLife* at this stage.

All the reviewers found your study to be of high quality, interesting and potentially suitable for publication in *eLife*. The reviewers especially valued the novelty of the manuscript demonstrating the role of COL17 in the homeostasis of the interfollicular epidermis. The data linking loss of COL17 to hyperproliferation and altered Wnt signalling were considered to be important and of interest. In addition, the link between aging and reduced COL17 level, resulting in hyperproliferation similar to the COL17 loss phenotype, were appreciated by the reviewers as they provide novel insights into understanding epidermal changes during skin aging.

However, there were some major concerns that need to be addressed and given their extent we found it unlikely that these can be addressed in the 2 month time window allowed for revisions in *eLife*. However, we would be happy to evaluate a new submission fully addressing the concerns of the reviewers. Based on the online discussion especially the following points are essential (but also other reviewer concerns should be considered).

The reviewers find that the link between WNT signalling and the phenotype are not solid enough and it needs to be validated in more detail:

Reviewer #2: The studies in Figure 2 showing that COL17 regulates neonatal IFE proliferation through Wnt signaling are descriptive. The involvement of Wnt signaling in COL17-deficient proliferative IFE phenotype needs to be demonstrated in vitro and/or in vivo" and Reviewer#1: The link between reduced Wnt signaling and hyperproliferation is surprising, as most of the literature associate Wnt as a pro proliferative signal in the epidermis (see for example (Lim et al., Science 2013). On the other hand Lef1 has been shown to be reduced as part of the skin inflammatory response (Quigley et al., Cell Reports 2016 PMID: 27425619). Given this, the causative link between Collagen XVII loss, suppression of Wnt and subsequent hyperproliferation should be more firmly demonstrated. Data showing that the K14-ΔNLef mouse has a similar phenotype is correlative at best, especially since it is not completely clear how this transgene regulates Wnt signaling. I realize that a rescue experiment with overexpression of a Wnt pathway component is time consuming, but perhaps an in vitro rescue or injection of a Gsk-3beta inhibitor"

Another major concern was related to the loss of the phenotype in the mice with age:

Reviewer#2: Figure 1 showed epidermal hyperproliferation was observed in the neonates of *Col17a1 -/-* mice. The authors need to explain and clarify why the proliferative IFE phenotype of *Col17a1 -/-* mice gradually weaned. In addition, the authors need to examine the skin of different regions to confirm if epidermis was hyperproliferative in region-specific manner."

Reviewer#3: Results, subsection “COL17 regulates neonatal IFE proliferation through Wnt-β-catenin signaling”: The authors find that the immunostaining of LEF1, β-catenin and PH3 are changed in the skin biopsies of two JEB patients with COL17A1 null mutations in a similar way to newborn *Col17a1-/-* mice. However, in the *Col17a1-/-* mice these changes are transient, but not in human. What is the explanation for this finding? Was there hyperproliferation in JEB epidermis?"

The Discussion was found to be two speculative and this need to be revised:

Reviewer#2: Discussion: The authors should discuss how their results are applied to human skin ie: What are the limitations of their study.

Discussion, paragraph six. I found this chapter highly speculative and I would suggest to re-write it. It is known that collagen XVII is involved in squamous cell carcinoma and recent papers have suggested its importance in colon and lung cancer (Moilanen et al., 2015, Liu et al., 2016 and 2016a). Moreover, I would not state that JEB patients with COL17A1 mutations have high frequency of multiple cutaneous SCC. In the paper by Yuen and Jonkman, only 2 out 14 JEB patients who developed SCC, were carriers of COL17A1 mutations. In addition, in the US National EB registry, the frequency of SCC in non-Herlitz JEB patients was very low (Fine et al., 2009).

Reviewer#3: The authors stated the possible role of COL17 in the prevention of oncogenesis in the Discussion but didn't provide any evidence for that. The authors need to study it in detail or tone down the statements.

Reviewer #1 had concerns regarding the statistics.

Reviewer #1:

This is an interesting manuscript on the role of type XVII in the interfollicular epidermis. The authors show that deletion of Col XVII results in a transient hyperproliferative responsem which the authors attribute to reduced cell adhesion and attenuated Wnt signaling. They further show that Col XVII is reduced during aging, resulting in hyperproliferation.

There are several interesting elements in the manuscript and the data is overall of good quality. The causative links between the observations are, however, not completely clear and should in my opinion be more firmly established prior to publication.

1) Collagen XVII is a critical adhesion molecule in the epidermis, highlighted by its involvement in human blistering disease. The authors should assess potential blistering and basement membrane integrity in there model in more detail by higher resolution imaging and preferably also EM. They should further exclude presence of inflammation at sites of "reduced attachment" that is almost invariably associated with disruption of the basement membrane, as this could be the cause of a transient hyperproliferative response.

2) Can the hyperproliferation phenotype be detected in isolated keratinocytes? This would provide further proof for a cell autonomous signaling defect.

3) The link between reduced Wnt signaling and hyperproliferation is surprising, as most of the literature associate Wnt as a pro proliferative signal in the epidermis (see for example (Lim et al., 2013). On the other hand Lef1 has been shown to be reduced as part of the skin inflammatory response (Quigley et al., Cell Reports 2016 PMID: 27425619). Given this, the causative link between Collagen XVII loss, suppression of Wnt and subsequent hyperproliferation should be more firmly demonstrated. Data showing that the K14-ΔNLef mouse has a similar phenotype is correlative at best, especially since it is not completely clear how this transgene regulates Wnt signaling. I realize that a rescue experiment with overexpression of a Wnt pathway component is time consuming, but perhaps an in vitro rescue or injection of a Gsk-3beta inhibitor.

4) In their discussion, the authors attribute the discrepant outcomes of altered Wnt (hypo vs hyperproliferation) to different anatomical locations (paw versus back skin). This could be easily tested by analyzing paw skin in their mouse model.

5) As the authors themselves point out, several reports have shown epidermal thinning during aging and this seems also to be the case in human aging. The authors should more critically discuss this discrepancy in the Discussion section and also perhaps mention the critical role of extrinsic factors such as the hygiene status of the animal housing facility that might influence this phenotype.

6) The exact n number used for the quantifications should be included in the figure legends. In addition, the statistics seem inappropriate as it is not clear why a parametric test (Students-t) is used in cases where the data is unlikely to have equal variance and normal distribution (for example IF analyses of data from 3 mice). A more appropriate test such as the Mann-Whitney U test should be applied.

Reviewer #2:

The authors examined the role of COL17 in regulating IFE homeostasis. How aging influences the homeostasis of the IFE and its stem cells has been elusive. This manuscript tackles this important issue. The data are in most parts beautiful. The study will provide novel insights into understanding epidermal changes during skin aging. There are additional experiments and discussions the authors can add to improve the manuscript.

1) Figure 1 showed epidermal hyperproliferation was observed in the neonates of *Col17a1 -/-* mice. The authors need to explain and clarify why the proliferative IFE phenotype of *Col17a1 -/-* mice gradually weaned. In addition, the authors need to examine the skin of different regions to confirm if epidermis was hypeproliferative in region-specific manner.

2) The studies in Figure 2 showing that COL17 regulates neonatal IFE proliferation through Wnt signaling are descriptive. The involvement of Wnt signaling in COL17-deficient proliferative IFE phenotype needs to be demonstrated in vitro and/or in vivo.

3) Interestingly, the treatment of young mouse IFE samples with PKC inhibitor diminished apico-lateral membraneous COL17. To confirm if the changes of COL17 distribution were α-PKC-dependent, the authors need to confirm the level of phospho-PKC α in the IFE samples treated with Go6983 and myr PSI.

4) The authors stated the possible role of COL17 in the prevention of oncogenesis in the discussion but didn't provide any evidence for that. The authors need to study it in detail or tone down the statements.

Reviewer #3:

In this manuscript the authors present convincing data that suggest a novel role to transmembrane collagen XVII to regulate the proliferation of interfollicular epidermis (IFE) in murine skin. They show that the lack of collagen XVII leads to transient IFE proliferation in the paw skin of neonatal *Col17a1-/-* mice. Analyzes of signaling pathways reveals that the IFE proliferation is mediated by the suppression of Wnt signaling. Accordingly, the immunostaining of LEF1 and β-catenin are decreased in basal cells of *Col17a1-/-* mice epidermis and also in skin biopsies obtained from newborn and 15-year-old JEB patient who carry COL17A1 null mutations. Transgenic rescue by the expression of human collagen XVII in *Col17a1-/-* mice abrogates IFE hypeproliferation and altered Wnt signaling. They further demonstrate that physical aging leads to IFE hyperproliferation in mice and at the same time, the amount of non-hemidesmosomal collagen XVII is reduced in both mice and human skin. Moreover, physical aging related changes of the calcium concentration of keratinocytes and the reduction of atypical protein kinase C (αPKC) contribute to the distribution of collagen XVII in aging IEF. Finally, overexpression of human collagen XVII suppresses the epidermal hyperproliferation with physical aging of IFE.

The study is of a great interest since it suggest a novel function of collagen XVII as an anti-aging molecule of the skin. However, I have some comments addressing the methodological aspects and interpretation of results.

1) Introduction, paragraph three: I would not call lamina lucida and lamina densa as components of basement membrane, since especially lamina lucida is generally considered as an artifact of tissue dehydration.

2) Introduction: Please add a recent study which suggests that the expression of collagen XVII is decreased in aged human skin (Langton et al., 2016). Please also discuss this paper.

3) Results, first paragraph: For readers who are not familiar with cutaneous biology, the authors could clarify the relationship between the murine paw skin and human skin.

4) Results, second paragraph: Why the expression of integrin β 2 and the three chains of laminin 332 were not measured?

5) Results, subsection “COL17 regulates neonatal IFE proliferation through Wnt-β-catenin signaling “: The authors find that the immunostaining of LEF1, ββ-catenin and PH3 are changed in the skin biopsies of two JEB patients with COL17A1 null mutations in a similar way to newborn *Col17a1-/-* mice. However, in the *Col17a1-/-* mice these changes are transient, but not in human. What is the explanation for this finding? Was there hyperproliferation in JEB epidermis?

6) Results, subsection “Distribution of COL17 is altered with physical aging”: Is there hyperproliferation of epidermis in aging human skin?

7) Results, Subsection “Atypical PKC controls the distribution of COL17 in IFE”: Does the treatment of skin samples with EDTA really simulate aged IFE or rather the changes of calcium concentration in aged skin?

8) Discussion. The authors should discuss how their results are applied to human skin ie; What are the limitations of their study.

9) Discussion, paragraph six. I found this chapter highly speculative and I would suggest to re-write it. It is known that collagen XVII is involved in squamous cell carcinoma and recent papers have suggested its importance in colon and lung cancer (Moilanen et al.,2015, Liu et al., 2016 and 2016a). Moreover, I would not state that JEB patients with COL17A1 mutations have high frequency of multiple cutaneous SCC. In the paper by Yuen and Jonkman, only 2 out 14 JEB patients who developed SCC, were carriers of COL17A1 mutations. In addition, in the US National EB registry, the frequency of SCC in non-Herlitz JEB patients was very low (Fine et al., 2009).

[Editors’ note: what now follows is the decision letter after the authors submitted for further consideration.]

Thank you for resubmitting your work entitled "Type XVII collagen coordinates proliferation in the interfollicular epidermis" for further consideration at *eLife*. Your revised article has been favorably evaluated by Anna Akhmanova (Senior editor), a Reviewing editor, and two of the original three reviewers.

The manuscript has been improved but there are some remaining issues that need to be addressed before acceptance, as outlined below:

The manuscript has improved as the link between col17 and Wnt has been better established (Wnt inhibitors, Top-gal), although especially the Top gal images don't look completely convincing regarding the selection of the area that is being measured. It would be better to have an unbiased approach for this.

However, at the same time the manuscript has become somewhat confusing: on one hand luciferase assays demonstrate that the Wnt defect is maintained in cell culture, on the other hand the proliferation phenotype is not present. What is the explanation for this? In addition, on one hand there are similarities in human and mouse skin but the keratinocytes show an opposite effect regarding proliferation.

In addition to this fundamental issue, many of the discrepancies in the data are dismissed by discussion instead of attempting to tackle them experimentally.

If the epidermal structure (and not a minor barrier defect) is really the underlying factor, analysis of organotypic raft cultures should recapitulate the phenotype. Alternatively, the authors are welcome to consider other experimental approaches to address this.

---

## [Author Response]

[Editors’ note: the author responses to the first round of peer review follow.]

All the reviewers found your study to be of high quality, interesting and potentially suitable for publication in eLife. The reviewers especially valued the novelty of the manuscript demonstrating the role of COL17 in the homeostasis of the interfollicular epidermis. The data linking loss of COL17 to hyperproliferation and altered Wnt signalling were considered to be important and of interest. In addition, the link between aging and reduced COL17 level, resulting in hyperproliferation similar to the COL17 loss phenotype, were appreciated by the reviewers as they provide novel insights into understanding epidermal changes during skin aging.

However, there were some major concerns that need to be addressed and given their extent we found it unlikely that these can be addressed in the 2 month time window allowed for revisions in eLife. However, we would be happy to evaluate a new submission fully addressing the concerns of the reviewers. Based on the online discussion especially the following points are essential (but also other reviewer concerns should be considered).

The reviewers find that the link between WNT signalling and the phenotype are not solid enough and it needs to be validated in more detail:

Reviewer #2: The studies in Figure 2 showing that COL17 regulates neonatal IFE proliferation through Wnt signaling are descriptive. The involvement of Wnt signaling in COL17-deficient proliferative IFE phenotype needs to be demonstrated in vitro and/or in vivo" and Reviewer#1: The link between reduced Wnt signaling and hyperproliferation is surprising, as most of the literature associate Wnt as a pro proliferative signal in the epidermis (see for example (Lim et al., 2013). On the other hand Lef1 has been shown to be reduced as part of the skin inflammatory response (Quigley et al., Cell Reports 2016 PMID: 27425619). Given this, the causative link between Collagen XVII loss, suppression of Wnt and subsequent hyperproliferation should be more firmly demonstrated. Data showing that the K14-ΔNLef mouse has a similar phenotype is correlative at best, especially since it is not completely clear how this transgene regulates Wnt signaling. I realize that a rescue experiment with overexpression of a Wnt pathway component is time consuming, but perhaps an in vitro rescue or injection of a Gsk-3beta inhibitor"

To further validate the relationship between Wnt signaling and COL17, we performed three key experiments (Figure 2) as follows:

a) The modified TOPFLASH reporter assay using HEK293 cells (Tsukiyama et al., 2015) revealed that COL17 expression stabilizes Wnt activity in vitro (Figure 2).

b) Crossing of ins-TOPGAL mice (a Wnt reporter animal) with *Col17a1*-/- showed that the absence of COL17 diminishes Wnt activity in paw IFE cells in vivo (Figure 2).

c) Intraperitoneal administration of two Wnt inhibitors (IWP-2 and Wnt-C59) into neonatal mice resulted in paw IFE hyperproliferation in P1 mice (Figure 2), which reproduced the result of the deltaNLef mice (Figure 2). This is contrasted by the IFE hypotrophy in the paw of 1-month-old mice with inducible loss-of-function of β-catenin (Lim et al., 2013). We believe that this discrepancy is due to the differences of observational time points (neonate vs 1-month-old) and/or in the systems used in each study (conventional vs inducible loss-of-function).

Taken together, these results strongly corroborate the substantial link between COL17 and Wnt signaling associated with paw IFE proliferation at P1.

Another major concern was related to the loss of the phenotype in the mice with age:

Reviewer#2: Figure 1 showed epidermal hyperproliferation was observed in the neonates of Col17a1 -/- mice. The authors need to explain and clarify why the proliferative IFE phenotype of Col17a1 -/- mice gradually weaned. In addition, the authors need to examine the skin of different regions to confirm if epidermis was hyperproliferative in region-specific manner."

Our mathematical models of epidermal development have predicted that, upon the loosening of the attachment of committed progenitor cells to the basement membrane zone, the epidermal thickness was transiently increased and gradually returned to the baseline (Figure 1), which was compatible with our observation of Col17a1-/- neonatal epidermis and with the similar transient hypertrophy of Itga6-/- and Itgb1-/- epidermis (Brakebusch et al., 2000; Niculescu C et al., 2011). We believe that transient epidermal hyperproliferation with gradual weaning is at least partially explained by the loosening of dermo-epidermal adhesion at the neonatal stage. In addition, our observations of deltaNLef (Figure 2) and Wnt-inhibitor treated mice (Figure 2) showed that inactive Wnt signaling leads to IFE proliferation in the paw skin of P1 mice. As the number of LEF1-positive cells in the *Col17a1-/-* paw IFE was comparable with that of control mice by P4, we reasoned that the effect of inactive Wnt signaling on epidermal proliferation in *Col17a1-/-* epidermis at P1 is reset at P4, although the detailed mechanisms of this transient effect remain elusive. We included this notion in the revised manuscript.

Quantification of proliferation markers in the back skin IFE from P1 *Col17a1-/-* and control mice showed no significant differences (Figure 1—figure supplement 2), confirming the site specificity of these cell dynamics. We suspect that this discordance between the paw epidermis and back skin IFE might be explained either by the influence of hair follicle development on back skin IFE at P1 (generally in stage 5) or by the distinct regulation of IFE at each location in the body (Rompolas et al., 2016, Roy et al., 2016, Sada et al., 2016). We have included this discussion in the revised manuscript.

*Reviewer#3: Results, subsection “COL17 regulates neonatal IFE proliferation through Wnt-β-catenin signaling”: The authors find that the immunostaining of LEF1, β-catenin and PH3 are changed in the skin biopsies of two JEB patients with COL17A1 null mutations in a similar way to newborn Col17a1-/- mice. However, in the Col17a1-/- mice these changes are transient, but not in human. What is the explanation for this finding? Was there hyperproliferation in JEB epidermis?"*

We admit that the sample size of JEB patients is too small to draw a definitive conclusion. However, as JEB patients with COL17A1 mutations are very rare and it is difficult to obtain samples from these patients, we cannot increase the sample size. We have no data on why the changes seen in *Col17a1-/-* mice are transient but are not in human JEB skin. We have added this limitation to the revised manuscript and moved the JEB patient data from the main figures to the supporting materials (Figure 2—figure supplement 2).

The Discussion was found to be two speculative and this need to be revised:

Reviewer#2: Discussion: The authors should discuss how their results are applied to human skin ie: What are the limitations of their study.

In light of chronic stimuli and non-UV exposure, it would be reasonable to state that the murine paw IFE may be equivalent to the human palmoplantar and buttock epidermis. We have included this information in the revised manuscript.

In addition, we have listed the limitations of our study in the revised manuscript as follows:

a) Chronic stimulation through ambulation might affect paw IFE homeostasis with physical aging.

b) Drug-induced aPKC inhibition on the epidermis may not completely exclude non-specific pharmacological effects.

c) Our strategy did not utilize the inducible expression or deletion of COL17.

Discussion, paragraph six. I found this chapter highly speculative and I would suggest to re-write it. It is known that collagen XVII is involved in squamous cell carcinoma and recent papers have suggested its importance in colon and lung cancer (Moilanen et al., 2015, Liu et al., 2016 and 2016a). Moreover, I would not state that JEB patients with COL17A1 mutations have high frequency of multiple cutaneous SCC. In the paper by Yuen and Jonkman, only 2 out 14 JEB patients who developed SCC, were carriers of COL17A1 mutations. In addition, in the US National EB registry, the frequency of SCC in non-Herlitz JEB patients was very low (Fine et al., 2009).

Reviewer#3: The authors stated the possible role of COL17 in the prevention of oncogenesis in the discussion but didn't provide any evidence for that. The authors need to study it in detail or tone down the statements.

We have toned down and extensively rewritten this section in the Discussion, as well as cited and discussed the papers according to the reviewer’s suggestion. In addition, we have deleted the statement that JEB patients with COL17A1 mutations have a high frequency of multiple cutaneous SCC in the revised manuscript.

Reviewer #1 had concerns regarding the statistics.

In accordance with the reviewer’s suggestion, we included the exact n number used for each experiment in the figure legends. In the revised manuscript, we also applied the Mann-Whitney test to the data, which were unlikely to have equal distribution (e.g., relative fluorescent intensity).

Reviewer #1:

This is an interesting manuscript on the role of type XVII in the interfollicular epidermis. The authors show that deletion of Col XVII results in a transient hyperproliferative responsem which the authors attribute to reduced cell adhesion and attenuated Wnt signaling. They further show that Col XVII is reduced during aging, resulting in hyperproliferation.

There are several interesting elements in the manuscript and the data is overall of good quality. The causative links between the observations are, however, not completely clear and should in my opinion be more firmly established prior to publication.

1) Collagen XVII is a critical adhesion molecule in the epidermis, highlighted by its involvement in human blistering disease. The authors should assess potential blistering and basement membrane integrity in there model in more detail by higher resolution imaging and preferably also EM. They should further exclude presence of inflammation at sites of "reduced attachment" that is almost invariably associated with disruption of the basement membrane, as this could be the cause of a transient hyperproliferative response.

EM images of *Col17a1-/-* mouse back skin were published by our group (Nishie et al., 2007). According to the reviewer’s suggestion, we examined the paw skin of P1 *Col17a1-/-* mice using EM (Figure 1—figure supplement 1). The result showed hypoplastic hemidesmosomes and blurred anchoring filaments, as reported in the back skin, suggesting disturbed integrity of the basement membrane zone in the *Col17a1*-/- paw epidermis.

We confirmed that there were no significant differences in the number of dermal immune infiltrates (i.e., T-cells, macrophages and neutrophils) in the paw skin between *Col17a1-/-* mice and control mice (Figure 1—figure supplement 1), thus excluding the involvement of inflammatory processes in the disorganization of the basement membrane zone in *Col17a1*-/- skin.

2) Can the hyperproliferation phenotype be detected in isolated keratinocytes? This would provide further proof for a cell autonomous signaling defect.

The isolated keratinocytes from *Col17a1-/-* neonatal epidermis from the whole body were shown to be less proliferative in the colony formation assay (Tanimura et al., 2011) as opposed to the in vivo IFE hyperproliferation of keratinocytes from *Col17a1*-/- neonates. To confirm the previous data, we tested several proliferation assays on normal human epidermal keratinocytes (NHEKs) treated with *COL17A1* siRNAs (Figure 1—figure supplement 2). These assays revealed that the proliferative potential of NHEKs treated with *COL17A1* siRNAs is not significantly different from that of the control cells. These data suggest that the hyperproliferation of the *Col17a1*-/- paw IFE in P1 mice is dependent on in vivo conditions.

3) The link between reduced Wnt signaling and hyperproliferation is surprising, as most of the literature associate Wnt as a pro proliferative signal in the epidermis (see for example (Lim et al., 2013). On the other hand Lef1 has been shown to be reduced as part of the skin inflammatory response (Quigley et al., Cell Reports 2016 PMID: 27425619). Given this, the causative link between Collagen XVII loss, suppression of Wnt and subsequent hyperproliferation should be more firmly demonstrated. Data showing that the K14-ΔNLef mouse has a similar phenotype is correlative at best, especially since it is not completely clear how this transgene regulates Wnt signaling. I realize that a rescue experiment with overexpression of a Wnt pathway component is time consuming, but perhaps an in vitro rescue or injection of a Gsk-3beta inhibitor.

Please see the response to the Editor.

4) In their Discussion, the authors attribute the discrepant outcomes of altered Wnt (hypo vs hyperproliferation) to different anatomical locations (paw versus back skin). This could be easily tested by analyzing paw skin in their mouse model.

Please see the response to the Editor.

5) As the authors themselves point out, several reports have shown epidermal thinning during aging and this seems also to be the case in human aging. The authors should more critically discuss this discrepancy in the Discussion section and also perhaps mention the critical role of extrinsic factors such as the hygiene status of the animal housing facility that might influence this phenotype.

To explain the discrepancy in the epidermal growth rate with aging (hyperproliferation in our studies and others (Stern and Bickenbach, 2007; Charruyer et al., 2009) vs hypoproliferation (Giangreco, Qin, Pintar and Watt, 2008)), we propose the following confounding factors (including the hygiene status as the reviewer noted) and discussed these factors in the revised manuscript:

a) UV exposure (which inhibits cell growth) on human skin.

b) A wide divergence of epidermal thickness among different locations on the body (Porter et al., 1998) and among human individuals (Waller and Maibach, 2005).

c) Differences in the hygiene status among animal facilities.

d) The influence of the development and growth cycles of hair on IFE of haired skin.

We believe that our observations can exclude the involvement of most of these extrinsic and intrinsic confounding factors by restricting our experiments to the paw IFE of congenic mice at one animal facility.

6) The exact n number used for the quantifications should be included in the figure legends. In addition, the statistics seem inappropriate as it is not clear why a parametric test (Students-t) is used in cases where the data is unlikely to have equal variance and normal distribution (for example IF analyses of data from 3 mice). A more appropriate test such as the Mann-Whitney U test should be applied.

Please see the response to the Editor.

Reviewer #2:

The authors examined the role of COL17 in regulating IFE homeostasis. How aging influences the homeostasis of the IFE and its stem cells has been elusive. This manuscript tackles this important issue. The data are in most parts beautiful. The study will provide novel insights into understanding epidermal changes during skin aging. There are additional experiments and discussions the authors can add to improve the manuscript.

1) Figure 1 showed epidermal hyperproliferation was observed in the neonates of Col17a1 -/- mice. The authors need to explain and clarify why the proliferative IFE phenotype of Col17a1 -/- mice gradually weaned. In addition, the authors need to examine the skin of different regions to confirm if epidermis was hypeproliferative in region-specific manner.

Please see the response to the Editor.

2) The studies in Figure 2 showing that COL17 regulates neonatal IFE proliferation through Wnt signaling are descriptive. The involvement of Wnt signaling in COL17-deficient proliferative IFE phenotype needs to be demonstrated in vitro and/or in vivo.

Please see the response to the Editor.

3) Interestingly, the treatment of young mouse IFE samples with PKC inhibitor diminished apico-lateral membraneous COL17. To confirm if the changes of COL17 distribution were α-PKC-dependent, the authors need to confirm the level of phospho-PKC α in the IFE samples treated with Go6983 and myr PSI.

We utilized two inhibitors to exclude the non-specific drug activities on the treated cells at the effective in vivo and in vitro concentrations (Atwood et al., 2013; Wu et al., 2006). However, we regret that we have so far failed to clearly show decreased aPKC activity in skin treated with these inhibitors, but we strongly believe that the alteration of COL17 dynamics, as a membranous protein, in treated epidermis alone manifests as aberrant cell polarity. To further confirm our results, genetically modified mice with inducible aPKC inactivation might help. We have discussed this limitation in the revised manuscript.

4) The authors stated the possible role of COL17 in the prevention of oncogenesis in the discussion but didn't provide any evidence for that. The authors need to study it in detail or tone down the statements.

Please see the response to the Editor.

Reviewer #3:

[…]

The study is of a great interest since it suggest a novel function of collagen XVII as an anti-aging molecule of the skin. However, I have some comments addressing the methodological aspects and interpretation of results.

1) Introduction, paragraph three: I would not call lamina lucida and lamina densa as components of basement membrane, since especially lamina lucida is generally considered as an artifact of tissue dehydration.

According to the reviewer’s suggestion, we have deleted this sentence from the revised manuscript.

2) Introduction: Please add a recent study which suggests that the expression of collagen XVII is decreased in aged human skin (Langton et al., 2016). Please also discuss this paper.

In accordance with the reviewer’s suggestion, we have cited and discussed this paper in the revised manuscript.

3) Results, first paragraph: For readers who are not familiar with cutaneous biology, the authors could clarify the relationship between the murine paw skin and human skin.

Please see the response to the Editor.

4) Results, second paragraph: Why the expression of integrin β 2 and the three chains of laminin 332 were not measured?

We suspect that the reviewer meant integrin β 4 rather than β 2. We examined the relative expression of *Itgb4, Lamb3* and *Lamc2* in *Col17a1-/-* mice and control mice (Figure 1—figure supplement 1). There were no significant differences in expression between the groups.

5) Results, subsection “COL17 regulates neonatal IFE proliferation through Wnt-β-catenin signaling “: The authors find that the immunostaining of LEF1, ββ-catenin and PH3 are changed in the skin biopsies of two JEB patients with COL17A1 null mutations in a similar way to newborn Col17a1-/- mice. However, in the Col17a1-/- mice these changes are transient, but not in human. What is the explanation for this finding? Was there hyperproliferation in JEB epidermis?

Please see the response to the Editor.

6) Results, subsection “Distribution of COL17 is altered with physical aging”: Is there hyperproliferation of epidermis in aging human skin?

According to a previous report (Waller et al., 2005), the proliferation rates and thickness of the epidermis depend on the body location in humans. We examined the expression of the proliferation markers in the planter skin of aged and young individuals by staining Ki-67 and PCNA. However, the number of marker-positive cells was highly variable even among a specific group. We suspect that the direction of sectioning relative to fingerprints greatly affected the results. In addition, human plantar skin is very large, and its thickness differs from the toe to the heel. As we can only utilize the available clinical samples, it is impossible to control the direction of sectioning in formalin-fixed paraffin-embedded samples and to ensure that the collected samples are derived from the exact anatomical area. Therefore, we are afraid that we cannot provide reliable data for this experiment.

7) Results, Subsection “Atypical PKC controls the distribution of COL17 in IFE”: Does the treatment of skin samples with EDTA really simulate aged IFE or rather the changes of calcium concentration in aged skin?

EDTA treatment simulates the changes of calcium concentration in the aged skin. We have accordingly corrected this in the manuscript.

8) Discussion. The authors should discuss how their results are applied to human skin ie: What are the limitations of their study.

Please see the response to the Editor.

9) Discussion, paragraph six. I found this chapter highly speculative and I would suggest to re-write it. It is known that collagen XVII is involved in squamous cell carcinoma and recent papers have suggested its importance in colon and lung cancer (Moilanen et al., Human Pathology; 2015, Liu et al., 2016 and 2016a). Moreover, I would not state that JEB patients with COL17A1 mutations have high frequency of multiple cutaneous SCC. In the paper by Yuen and Jonkman, only 2 out 14 JEB patients who developed SCC, were carriers of COL17A1 mutations. In addition, in the US National EB registry, the frequency of SCC in non-Herlitz JEB patients was very low (Fine et al., 2009).

Please see the response to the Editor.

[Editors' note: the author responses to the re-review follow.]

The manuscript has improved as the link between col17 and Wnt has been better established (Wnt inhibitors, Top-gal), although especially the Top gal images don't look completely convincing regarding the selection of the area that is being measured. It would be better to have an unbiased approach for this.

As described in the original manuscript (Materials and method section), neonatal paw skin partially contains hair follicles and sweat glands, which are the major sources of Wnt activities. So, we definitely needed to exclude the effects from those appendages to adequately assess the relationship between IFE and Wnt signaling. Our observational area is confined to the regions that do not feature either hair follicles or sweat glands. We believe that this methodology is unbiased in terms of evaluating IFE in hindpaw skin.

To clarify our methodology, we have provided details on the localization of those appendages in paw skin (Figure 2—figure supplement 2). In this supplementary figure, we clearly indicated the area for the quantification for LacZ staining of ins-TOPGAL mice. We added the low and high magnification photos of these stainings in the figure (Figure 2—figure supplement 2).

However, at the same time the manuscript has become somewhat confusing: on one hand luciferase assays demonstrate that the Wnt defect is maintained in cell culture, on the other hand the proliferation phenotype is not present. What is the explanation for this?

The TOPFLASH assay in our experiments did not indicate the defect of Wnt signaling in the absence of COL17. Instead, the assay showed that overexpression of human COL17 stabilizes and amplifies the Wnt signaling in vitro. These data confirm the intrinsic interaction between COL17 and Wnt signaling using HEK293 cells rather than addressing proliferation dynamics in IFE keratinocytes. These data may support the fact that Wnt ligands, known as hydrophobic, are stabilized with extracellular matrix (Rozario T and DeSimone DW, Dev Biol 2010; Willert K and Nusse R, Cold Spring Harb Perspect Biol 2012). We will discuss the discrepancy between in vitro cultured keratinocytes and in vivo *Col17a1-/-* paw IFE skin in point 4 below.

In addition, on one hand there are similarities in human and mouse skin but the keratinocytes show an opposite effect regarding proliferation.

The proliferation assays using normal human epidermal keratinocytes with COL17 knockdown showed a slight reduction in proliferation, and murine keratinocytes obtained from *Col17a1-/-* mice were reported to exhibit a similar reduction in proliferative ability (Tanimura et al., 2011). Based on these results, we believe that human and murine cultured keratinocytes with COL17 deficiency show similar characteristics in proliferation. We added this information to the revised manuscript and cited the reference mentioned above.

In addition to this fundamental issue, many of the discrepancies in the data are dismissed by discussion instead of attempting to tackle them experimentally.

If the epidermal structure (and not a minor barrier defect) is really the underlying factor, analysis of organotypic raft cultures should recapitulate the phenotype. Alternatively, the authors are welcome to consider other experimental approaches to address this.

We performed COL17 knock down experiments on reconstituted 3D epidermis using LabCyte EPI-KIT (J-TEC, Aichi, Japan) in accordance with the provider’s protocol. In brief, we seeded human epidermal keratinocytes transfected with two COL17A1 siRNAs (Thermoscientific) or mock using Lipofectamine 3000 (Thermoscientific). The cells were cultured for 7days in cups in 24-well formats. The surface of the cells was exposed to the air interface for the cells to differentiate and reconstitute 3D epidermis. We treated these epidermis with 10µM BrdU (Σ) in the medium for 4hrs before the sampling. Utilizing qRT-PCR, we confirmed that COL17A1 gene expression was reduced in 3D epidermis treated with siRNAs (Figure 8). In contrast to the hyperproliferative phenotype of in vivo IFE paw skin in *Col17a1-/-* mice at P1, the 3D reconstituted epidermis treated with COL17A1 siRNA did not show a consistent proliferation tendency through PH3, BrdU and PCNA staining (Figure 8). Our observations on *Col17a1-/-* paw IFE showed that the phenotype starts from hyperproliferation in the neonatal stage but this tendency for proliferation is lost at 3 weeks-old. We believe that in vitro experiments on either 2D or 3D cultured cells may not reflect the chronological dynamics in in vivo epidermis (e.g., we cannot tell whether cultured cells in vitro correspond to 1 day-old or 20 day-old in vivo epidermis). Therefore, we do not include these data in the revised manuscript. We are aware that there are discrepancies between in vivo and in vitro data (2D or 3D) as the reviewers indicated. In other words, the hyperproliferative phenotype of *Col17a1-/-* paw IFE is simply dependent on in vivo conditions. We refined the text to describe these discrepancies in the revised manuscript.

Author response image 1.Proliferative ability of 3D reconstituted human epidermis treated with *COL17A1* siRNAs**DOI:**
http://dx.doi.org/10.7554/eLife.26635.022

(a) Quantitative RT-PCR (qRT-PCR) of COL17A1 gene (n=8).

(b) The number of epidermal basal cells positively labeled for PH3 per mm epidermis (Mock; n=5, siRNAs; n=8, respectively).

(c-d) The number of epidermal basal cells positively labeled for BrdU (c) and PCNA (d) per mm epidermis (n=8). The data are presented as the means ± SE. *0.01<p<0.05, ****p<0.0001., One-way ANOVA followed by Tukey's test.

</Figure 8 title/legend>